# The Identity of the Constriction Region of the Ribosomal Exit Tunnel Is Important to Maintain Gene Expression in *Escherichia coli*

Sarah B. Worthan,[a] Elizabeth A. Franklin,[a] Chi Pham,[a] Mee-Ngan F. Yap,[b] Luis R. Cruz-Vera[a]

[a]Department of Biological Sciences, University of Alabama in Huntsville, Huntsville, Alabama, USA
[b]Department of Microbiology-Immunology, Northwestern University Feinberg School of Medicine, Chicago, Illinois, USA

Sarah B. Worthan and Elizabeth A. Franklin contributed equally to this article. Author order was decided by the last effort performed during sending the manuscript to the journal.

**ABSTRACT** Mutational changes in bacterial ribosomes often affect gene expression and consequently cellular fitness. Understanding how mutant ribosomes disrupt global gene expression is critical to determining key genetic factors that affect bacterial survival. Here, we describe gene expression and phenotypic changes presented in *Escherichia coli* cells carrying an uL22(K90D) mutant ribosomal protein, which displayed alterations during growth. Ribosome profiling analyses revealed reduced expression of operons involved in catabolism, indole production, and lysine-dependent acid resistance. In general, translation initiation of proximal genes in several of these affected operons was substantially reduced. These reductions in expression were accompanied by increases in the expression of acid-induced membrane proteins and chaperones, the glutamate-decarboxylase regulon, and the autoinducer-2 metabolic regulon. In agreement with these changes, uL22 (K90D) mutant cells had higher glutamate decarboxylase activity, survived better in extremely acidic conditions, and generated more biofilm in static cultures compared to their parental strain. Our work demonstrates that a single mutation in a non-conserved residue of a ribosomal protein affects a substantial number of genes to alter pH resistance and the formation of biofilms.

**IMPORTANCE** All newly synthesized proteins must pass through a channel in the ribosome named the exit tunnel before emerging into the cytoplasm, membrane, and other compartments. The structural characteristics of the tunnel could govern protein folding and gene expression in a species-specific manner but how the identity of tunnel elements influences gene expression is less well-understood. Our global transcriptomics and translatome profiling demonstrate that a single substitution in a non-conserved amino acid of the *E. coli* tunnel protein uL22 has a profound impact on catabolism, cellular signaling, and acid resistance systems. Consequently, cells bearing the uL22 mutant ribosomes had an increased ability to survive acidic conditions and form biofilms. This work reveals a previously unrecognized link between tunnel identity and bacterial stress adaptation involving pH response and biofilm formation.

**KEYWORDS** acid resistance, biofilms, ribosomes, translational control

Address correspondence to Luis R. Cruz-Vera, luis.cruz-vera@uah.edu.

The authors declare no conflict of interest.

All actively translating nascent peptides must traverse the approximately 100 Å long ribosomal exit tunnel during their synthesis. The ribosomal exit tunnel is composed of both ribosomal proteins and rRNA residues which contribute to the architecture and irregular shape of the tunnel, resulting in a cavity that contains bumpy and pitted walls of varying widths (1). The narrowest region of the tunnel, named the constriction region, is comprised of the extended loops of the two riboproteins uL4 and

uL22 (1). The constriction region separates the tunnel into two segments, an upper and lower segment. The upper segment, located between the ribosome active site known as the peptidyl transferase center (PTC) and the constriction region, is about 35 Å long (1, 2) and can accommodate between 12 and 16 amino acid residues of a fully extended or alpha helical nascent peptide during translation (2). The upper segment of bacterial ribosomes is less variable than the lower segment (1) and is known to contain binding sites for antibiotics (3), small metabolites, and nascent peptides that are involved in regulating the progression of translation (2). Passing the constriction region, the lower segment of the tunnel widens, allowing nascent peptides to acquire initial folding structures (4). Due to its structural location, the constriction region is considered the gateway between the two functional upper and lower segments of the exit tunnel (5).

Changes in the components constituting the upper segment of the tunnel and constriction region affect the translational regulatory function of nascent peptides and the binding of small molecules and antibiotics (2, 3, 6). Consequences of these changes in the ribosome exit tunnel can affect bacterial physiology and behavior. Previously documented mutations in the extended loops of uL4 and uL22 have been shown to affect the functioning of regulatory nascent peptides that interact with small molecules in the exit tunnel, controlling protein synthesis and gene expression (2). An example of such mutational effects is seen during the regulation of the *tnaCAB* (*tna*) operon, which is responsible for the breakdown of L-tryptophan (L-Trp) to indole (7). The *tna* operon is controlled by a dedicated transcriptional attenuation mechanism that senses free L-Trp (7). Free L-Trp interacts at the constriction region of ribosomes translating the *tna* leader peptide (8), *tnaC*, inducing translation arrest and allowing the transcription of the downstream genes expressing tryptophanase (TnaA) and a L-Trp permease (TnaB) (7). Mutational changes in the uL22 K90 residue abolish L-Trp-induced translation arrest at the *tnaC* gene and consequently the expression of the *tna* operon and production of indole (9). Indole functions as an intracellular signal and regulates phenotypes critical to bacterial survival, such as cellular division (10), biofilm formation (11), acid resistance (12), antibiotic resistance (13), and chemotaxis (11). Well characterized uL4 and uL22 erythromycin resistant mutations have shown various global translational effects as well. A mutation in uL4 that changed a highly conserved residue produces multiple translation defects (14). A deletion in the extended loop of uL22 (Δ82-84) that widens the constriction region (15) affects expression of the *tna* operon, functioning of the bacterial Sec secretory system, and consequently the expression of several genes related to bacterial virulence and survival (16). Overall, these observations indicate that changes in the extended loops of uL4 and uL22 proteins affect translation progression, the global expression of genes, and cellular fitness (16). Furthermore, studies on regulatory nascent peptides have shown that sequence variations among uL22 homologs exhibit species-specific expression of genes (6). For example, the MifM translation regulatory nascent peptide, whose major function is to sense the activity of membrane insertase in *Bacillus subtilis*, shows dependence on the nature of the amino acid residue at the 90th position of uL22. The MifM protein does not display regulatory activity on *Escherichia coli* ribosomes until the *E. coli* uL22 K90 residue is changed to a methionine residue that is observed in *B. subtilis* uL22 protein (6). This suggests that variations in certain residues of the extended loop of uL22 observed among bacteria may be to optimize the needs of gene expression for each bacterial species. Therefore, single mutational changes in the extended loop of uL22 could alter global expression of genes in bacteria.

In this work, we set out to determine the effects of single residue changes in the non-conserved positions of the extended loops of uL4 and uL22 on global gene expression in *E. coli*. Initially, growth curves were conducted to assess the *in vitro* cellular fitness of cells harboring mutational changes in uL4 and uL22. The mutation uL22 (K90D) displayed a significant change in cell growth, making it an ideal candidate to determine the effect of this mutation on global gene expression. Ribosome profiling

analysis indicated alterations in levels of mRNA transcripts for approximately 7% of genes and changes in translation levels for 1% of genes expressed. We found reductions in ribosome protection at the beginning of several mRNAs that pinpoint possible candidate genes whose translation initiation or elongation could be affected directly by ribosomes harboring the uL22(K90D) mutational change. Notably, two genes related to the acid resistance response, *cadB* and *adiY*, are translationally attenuated. Genes related to the metabolism of alternative carbon sources and amino acids showed translational reductions, as well as reductions in their mRNA levels. Furthermore, these expressional changes may induce compensatory alterations in the expression of other genes. There were increases in the mRNA abundance of genes involved in the glutamate-dependent acid resistance response and in the uptake and modification of the autoinducer-2 (AI-2) signaling molecule. The differential expression of these genes is corroborated with an increased formation of biofilms and acid tolerance of uL22(K90D) cells.

## RESULTS

**Variability of the uL4 and uL22 amino acid residues that constitute the constriction region of the ribosomal exit tunnel.** Previous structural and computational studies of ribosomes indicate that specific residues of the extended loops of uL4 and uL22 are in the path of actively synthesized nascent chains (Fig. 1A) (2, 17). It has been suggested that interactions between these residues and nascent peptides possibly affect the rate of translation elongation/termination and support initial folding of nascent peptides (2, 18). Thus, these interactions potentially have the ability to modulate gene expression and protein production. Given that previous analyses have suggested the variability of uL4 and uL22 amino acid residues among bacteria is governed by the necessity for species-specific protein synthesis regulation (6) and because we were interested in understanding the role uL4 and uL22 play in global gene expression, we initially decided to determine the chemical conservation of the residues constituting the extended loops of these ribosomal proteins by comparing their orthologs among prokaryotic and eukaryotic organisms. For eubacterial species, we compared 231 sequences spanning 16 phyla using a Weblogo generator that creates sequence logos based on multisequence alignments and displays the conservational frequency of an amino acid at a given position (Fig. 1B). We noted that in eubacteria while many positions were highly conserved, likely due to important structural interactions or functions, others were more variable. Our analyses revealed that the extended loop sequences of both uL4 and uL22 contain an enrichment of basic amino acids and an absence of acidic residues (Fig. 1B), consistent with previous observations (1). We also observed that many of the conserved residue positions containing basic amino acids in these loops point inward toward rRNA. However, the majority of the residues pointing outward into the constriction region of the tunnel and toward the growing nascent peptide, are variable. High variability was observed in the 61st position of uL4 and the 90th position of uL22, with the majority of orthologous proteins containing basic amino acids in these residue positions. Low variability was observed in the 67th residue position in uL4 and the 92nd residue position in uL22 (Fig. 1B). Similar observations were noted for eukaryotic uL4 and uL22 residues. Although in eukaryotes the overall conservation of residues was more prominent compared with eubacteria, there was an analogous enrichment of basic amino acids, especially in residues which face the tunnel, with the exception of a glutamic acid residue at the 61st position of uL4 (Fig. 1B). Overall, our *in-silico* analyses indicated that in eubacteria the extended loop residues of uL4 and uL22 pointing toward the exit tunnel are frequently variable. However, despite their variability, there is a general avoidance of negatively-charged amino acids in these positions.

**Effects of replacing non-conserved amino acid residues of the uL4 and uL22 extended loops on cell growth.** To investigate the physiological significance of the noted avoidance of acidic amino acids at the variable extended loop residues of uL4 and uL22, we initially studied the growth fitness of bacteria expressing either uL4 or uL22 mutant proteins containing an aspartic acid in place of the native basic residue in

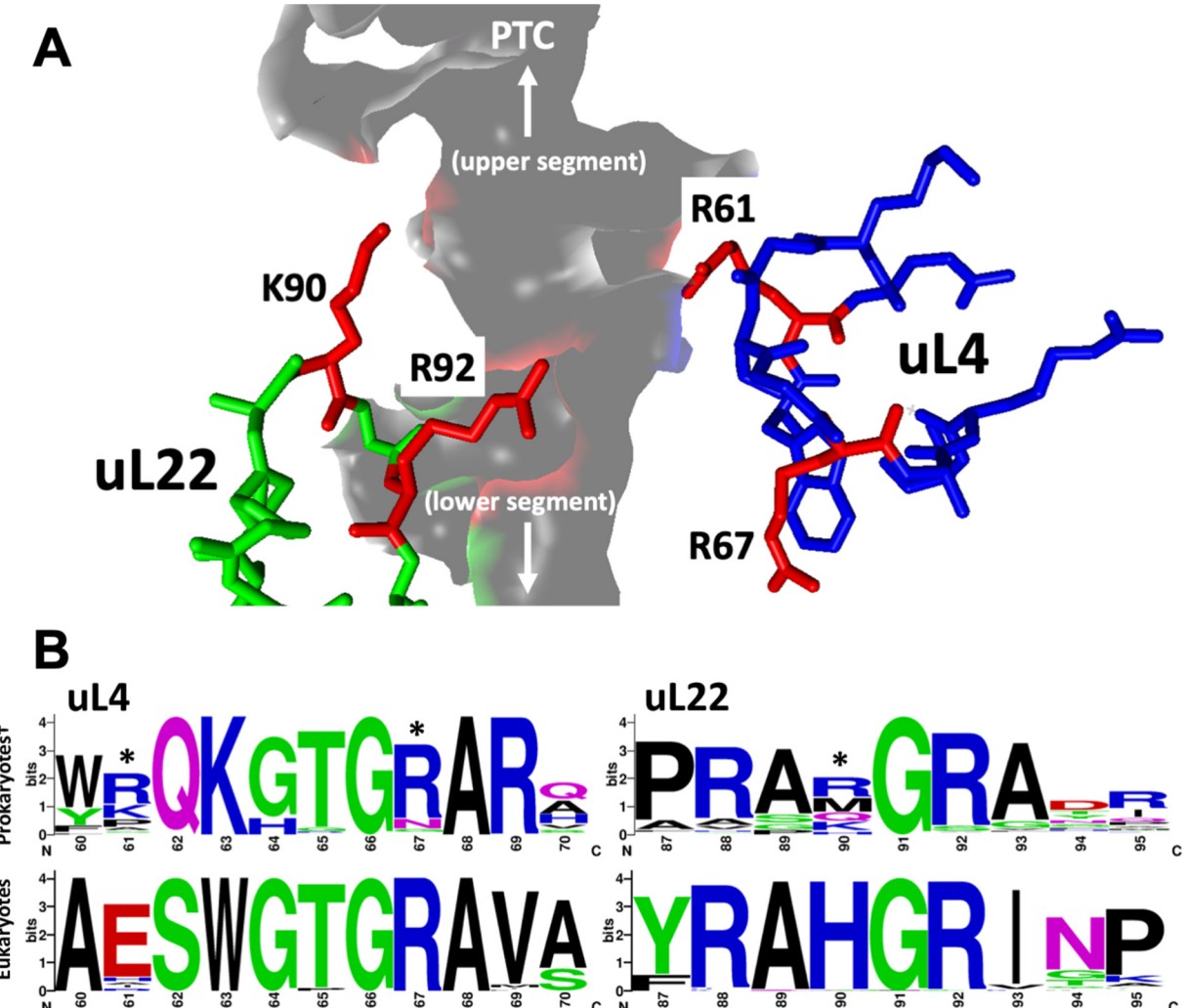

**FIG 1** Comparative analysis of the extended loop amino acid sequences of uL4 and uL22 from diverse prokaryotic and eukaryotic organisms. (A) Figure obtained from the PDB:4UY8 structure (65), shows the extended loops of ribosomal proteins uL4 (blue sticks) and uL22 (green sticks) surround the lumen of the ribosomal exit tunnel (gray surface). Non-conserved residues uL4 (R61 & R67) and uL22(K90 & R92), are shown in red. The orientation of the PTC and the tunnel exit are indicated with arrows. (B) Weblogo results of multiple protein alignment analyses of segments of the extended loops of uL4 and uL22 proteins that constitute the constriction region of the ribosomal exit tunnel. These four Weblogos represent the conservation and frequency of amino acid residues at specific positions (*E. coli* numbering). Prokaryotic sequences are shown on top (231 species) while eukaryotic sequences are shown on bottom (56 species). Residue positions studied in this work are marked with an asterisk. No archaea sequences were included in the alignments of prokaryotes.

rich media, Luria broth (LB). We did not observe a significant reduction in doubling times during the exponential growth phase in strains expressing the mutant proteins with respect to the strains expressing the wild-type proteins (Fig. 2A). However, we did observe that uL22(K90D) bacterial cultures consistently displayed a significant growth reduction compared with the uL22 wild-type bacterial cultures at the transition between the exponential and stationary growth phases (Fig. 2B, compare half-closed circles plot with open circles plot). Furthermore, the growth of the double mutant uL4(R61D)/uL22(K90D) behaved more similarly to the single mutant uL22(K90D) cells than the uL4(R61D) cells during this transition phase (Fig. 2, compare half-closed triangles plot with half-closed circles plot and open triangles plot). This latter observation suggested that the uL22(K90D) mutation had the largest impact on the detrimental effects in the cell growth of our bacterial strains. Therefore, the consistent reduction of growth at the transition between exponential and stationary phase of cells containing the uL22(K90D) mutant protein indicated a phenotypic change warranting further investigation.

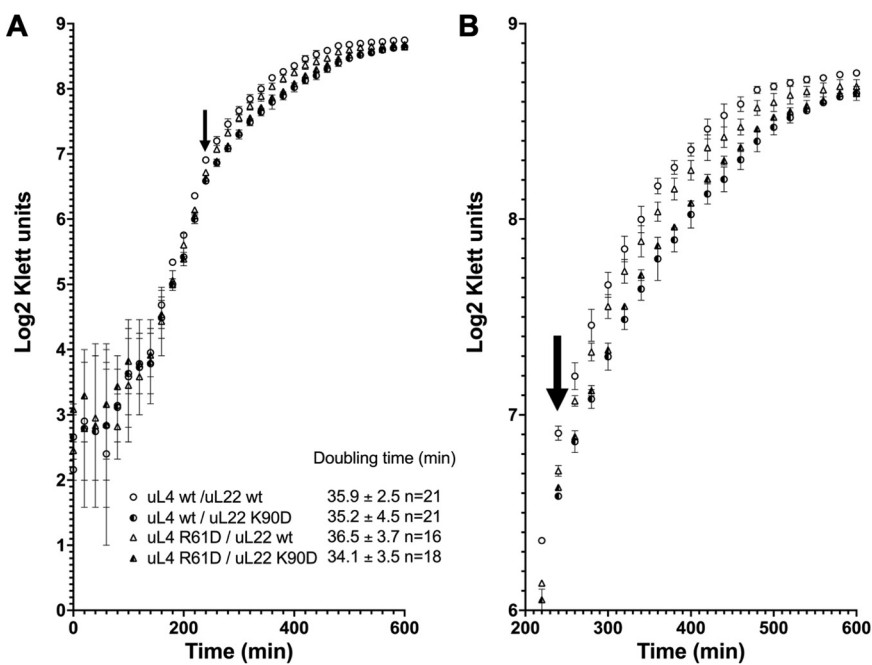

**FIG 2** Cell growth of wild type and ribosomal protein mutant strains in rich media. Plots representing the growth of strains expressing wild type (WT) uL4 and uL22 proteins and indicated uL4(R61D) or uL22 (K90D) mutant proteins. Growth is shown (A) over a 10-h period and (B) from 200 to 600 min to emphasize disparities in growth. *y* axes displayed Log$_2$ values of determined Klett units. Arrows indicate the time where cultures samples were obtained for ribosome profiling and RNA-seq analyses. Each curve shown is the resulting average of three independent experiments. Doubling times for each strain were calculated using data points between 140 min and 150 min and 240 min from several replicate growth curves as indicated in Materials and Methods. It should be noted that these cells replicate slowly which we suspect is because they contain several deletions used to analyze the expression of the *tnaC-tnaA-lacZ* reporter gene (Table S1).

**Global changes in gene expression observed in the uL22(K90D) mutant strain.** We next sought to determine differences in gene expression between wild-type and uL22(K90D) cells in efforts to provide insight into the variations in cell growth behavior. RNA-seq and ribosome profiling assays were performed on wild-type and uL22(K90D) cells grown in LB media to quantitate mRNA abundance and ribosome occupancy, respectively (see Materials and Methods). With the exception of translational stalling, ribosome occupancy is a strong proxy for translational efficiency. For these assays, cells were harvested after growing for 4 h, which reflected the time point when we began observing the separation in growth between the cultures (Fig. 2, arrows). Comparison of three biological replicates indicated high reproducibility between samples in both normalized RNA and normalized ribosome protected fragments (RPFs) read counts (Fig. S1A and S1B, respectively). This reproducibility was also corroborated by a principal-component analyses (PCA) plot with replicates separating according to the tested strains and harvesting dates (Fig. S1C). The RNA-seq and ribosome profiling data revealed that approximately 8.3% of the 3,729 analyzed genes in our uL22(K90D) mutant strain were affected, compared with its parental wild-type strain. Plotting RPFs versus mRNA levels indicated good correlation ($R^2 = 0.67$) between both values (Fig. 3A), exhibiting a distribution of points consistent with changes of mRNA levels as a major factor that affects translation and the expression of genes in the mutant bacteria (19). In the uL22(K90D) strain, approximately 280 genes showed reduction in their expression with respect to the wild-type strain (Fig. 3A). Meanwhile, 260 genes showed increases in their expression compared to the wild-type strain. Overall, the gene expression data indicate that the uL22(K90D) mutation alters both mRNA abundance and ribosome occupancy on mRNA in a subset of genes.

**Genes with reduced expression in the uL22(K90D) mutant strain.** In general, our data indicated that genes found in operons involved in pH regulation and carbon and

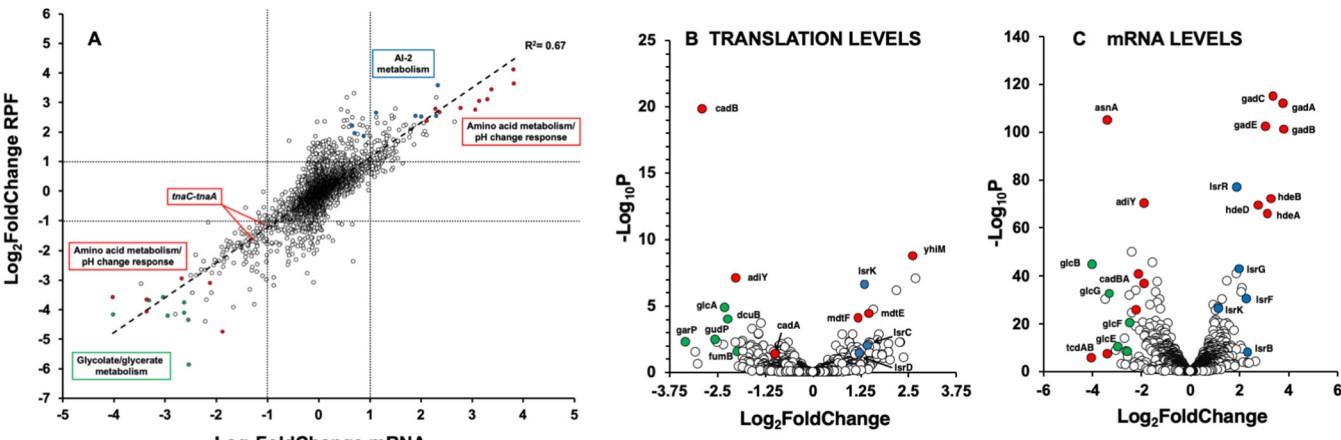

**FIG 3** Differential changes of translational and mRNA levels in the uL22(K90D) mutant strain. (A) Scatterplot showing Log$_2$ fold changes of RPF read counts versus their corresponding Log$_2$ fold changes of mRNA read counts per gene plotted. (B) Volcano plot showing significance versus changes in translation efficiency (RPF read counts/mRNA read counts) of all tested genes. (C) Volcano plot showing significance versus changes in mRNA read counts. Affected genes from the same operons and/or within the same cellular pathways are color-coded in green, blue, and red. Data to make above graphs can be found in supplemental file.

amino acid metabolism had a reduced expression in the uL22(K90D) mutant strain (Fig. 3A). By calculating translation efficiency (RPFs/mRNA levels) (see Materials and Methods), we determined that a small subset of these genes was in fact translationally affected. One of the most significantly translationally affected genes was the *cadB* gene, which is part of the *cadBA* operon involved in the lysine-dependent acid resistance system in *E. coli* (20). The translation efficiency of *cadB* was reduced approximately 7.5-fold in the uL22(K90D) mutant compared to the wild-type strain (Fig. 3B, Table 1), while its mRNAs levels were also reduced in the uL22(K90D) mutant about 4-fold (Fig. 3C, Table 1). This means that despite only a 4-fold reduction in mRNA levels, the number of ribosomes engaged with *cadB* mRNA molecules was 28-fold reduced in the uL22(K90D) mutant. Consequently, we also detected a significant reduction of ribosomes engaged at the start codon of *cadB* (Fig. 4A, asterisk), which corresponded with low mRNA concentrations in the same position of the gene (Fig. 4A). This observation suggests that the *cadB* gene may have low translation initiation efficiency in the uL22 (K90D) mutant strain or ribosomes translating the first codons frequently fall off the messenger. Additionally, we also observed significant reductions in the mRNA concentrations of *cadA* (Fig. 3C, Table 1), especially at its 3′-end (Fig. 4A), which suggests low transcriptional expression and/or high degradation of *cadBA* operon transcripts. The expression of genes within this operon is activated by several transcription factors, including GadE-RcsB, GadX, and CadC (21–23). We did not observe any significant reductions in the expression of the *cadC* gene (Fig. 4A) or *rcsB* gene (see supplemental file). Although, we did observe an increased expression of the *gadE* gene as discussed below (Fig. 4C, Table 2). Because the expression of known transcriptional regulators which control *cadBA* expression cannot explain low *cadBA* mRNA concentrations in uL22(K90D), we suspect that the low translation efficiency of *cadB* may affect the stability of the operon's mRNA, leading to rapid mRNA turnover. The gene *adiY* also displayed reduced translation efficiency in the uL22(K90D) strain (Fig. 3B, Table 1). Like *cadB*, *adiY* also plays a role in the acid response as part of the arginine-dependent acid resistance system and activates the expression of additional acid response genes, such as *gadA* and the *gadBC* operon (24). Similar to what was observed for *cadB*, *adiY* transcripts showed a reduction in ribosome occupancy at its start codon and low mRNA concentrations in the same position (Fig. 4B, asterisk). This suggests that the expression of *adiY*, like *cadB*, in the uL22(K90D) strain could be affected during translation initiation or during translation elongation of its first codons.

Several operons involved in amino acid metabolism showed reductions in expression and we noticed that modifications of these pathways were mostly related with

**TABLE 1** Fold reduction[a] in mRNA abundance, RPF, and translation efficiency for the expression of metabolic related genes affected in the uL22(K90D) strain

| Operons/genes | mRNA | RPFs | RPFs/mRNA levels |
|---|---|---|---|
| _glcDEFGBA_[c] | Glycolate catabolism | | |
| glcD | −5.9 ± 1.9 | **−20.6 ± 6.0** | **−3.6 ± 0.6**[b] |
| glcE | −7.7 ± 2.3 | **−18.4 ± 5.2** | **−2.5 ± 0.4** |
| glcF | −5.5 ± 2.1 | −10.8 ± 3.8 | −1.9 ± 0.3 |
| glcG | −10.0 ± 2.4 | −12.9 ± 3.3 | −1.3 ± 0.2 |
| glcB | −16.1 ± 3.9 | −18.1 ± 4.8 | −1.1 ± 0.1 |
| glcA | −1.8 ± 0.2 | **−9.1 ± 3.7** | **−4.7 ± 0.5** |
| garPLRK | Galactarate catabolism | | |
| garP | −5.8 ± 1.8 | **−57.8 ± 25.1** | **−9.0 ± 1.3** |
| garL | −6.2 ± 1.8 | **−17.4 ± 5.0** | **−2.6 ± 1.3** |
| garR | −2.9± 1.6 | −3.3 ± 2.1 | −1.2 ± 0.4 |
| garK | −2.6 ± 0.5 | −3.6 ± 1.0 | −1.4 ± 0.1 |
|  | Galactarate catabolism transcriptional regulator | | |
| cdaR | −3.8 ± 1.1 | −5.1 ± 1.3 | −1.3 ± 0.4 |
| dcuBfumB | Fumarate catabolism | | |
| dcuB | −4.6 ± 1.1 | **−21.1 ± 4.4** | **−4.6 ± 1.0** |
| fumB | −2.0 ± 0.3 | **−7.9 ± 5.0** | **−3.4 ± 0.5** |
| tdcABCDEFG | Threonine/serine catabolism | | |
| tdcA | −16.2 ± 6.3 | −12.0 ± 10.1 | −1.2 ± 0.6 |
| tdcB | −10.3 ± 4.6 | −16.9 ± 3.9 | −1.9 ± 0.4 |
| tdcC | −1.5 ± 0.2 | −3.0 ± 0.8 | −1.8 ± 0.4 |
| tdcD | −1.1 ± 0.1 | −2.2 ± 0.8 | −1.9 ± 0.6 |
| dcuAaspA | Aspartate catabolism | | |
| aspA | −2.9 ± 0.3 | −3.7 ± 1.0 | −1.3 ± 0.5 |
| dcuA | −2.9 ± 0.3 | −3.7 ± 1.2 | −1.3 ± 0.1 |
|  | Asparagine synthesis | | |
| asnA | −10.2 ± 0.9 | −12.6 ± 0.8 | −1.2 ± 0.3 |
| _cadBA_ | Lysine decarboxylation | | |
| cadB | −3.7 ± 0.6 | **−26.9 ± 4.8** | **−6.8 ± 2.7** |
| cadA | −4.3 ± 0.3 | **−8.5 ± 1.8** | **−2.0 ± 0.2** |
|  | Acid response transcriptional regulator | | |
| adiY | −2.5 ± 0.2 | **−10.0 ± 3.0** | **−4.0 ± 1.0** |

[a]Standard deviation calculations can be found in the supplemental file.
[b]Bold numbers indicate marked reduction of translation efficiency.
[c]The underline terms indicate the name of the operon which gene's values are shown in the tables.

the generation and consumption of ammonia. As expected, the *tnaC* and *tnaA* genes from the *tnaCAB* operon and the corresponding *tnaA-lacZ* protein fusion expressed in these cells (Table S1) displayed low mRNA levels and RPF values (Table 3). We have confirmed that production of the TnaA tryptophanase enzyme (Fig. S2A, B), as well as the *tnaA-lacZ* fusion protein, were reduced in the uL22(K90D) mutant strain (Fig. S2C). We also observed a reduction in the synthesis of indole in the uL22(K90D) mutant strain, confirming a reduction in the activity of the tryptophanase enzyme (Fig. S2D). We did not see a drastic difference (2-fold) in the *tnaA* mRNA levels between both strains (Table 3), in contrast to the ~10-fold difference seen at the protein levels (Fig. S2). We suspect that once the level of tryptophanase enzyme is elevated in the uL22(WT) strain, the concentrations of *tnaC* inducer (tryptophan) is decreased, thereby reducing the transcriptional induction of the *tnaA* mRNA (25), making the differences between both strains smaller than expected. Similar effects were also observed for the *tnaB* gene of the *tnaCAB* operon, which did not show significant changes in the mRNA and RPF values (Table 3). It is important to note that RNase P processes the intergenic region between *tnaA* and *tnaB* (26), resulting in the degradation of *tnaB* mRNA. RNA turnover may account for the low *tnaB* levels in comparison to that of the *tnaA* and *tnaC* mRNAs, resulting in even smaller difference between the strains. In addition to the *tnaA* gene, the *asnA* gene also exhibited a significant reduction in its expression

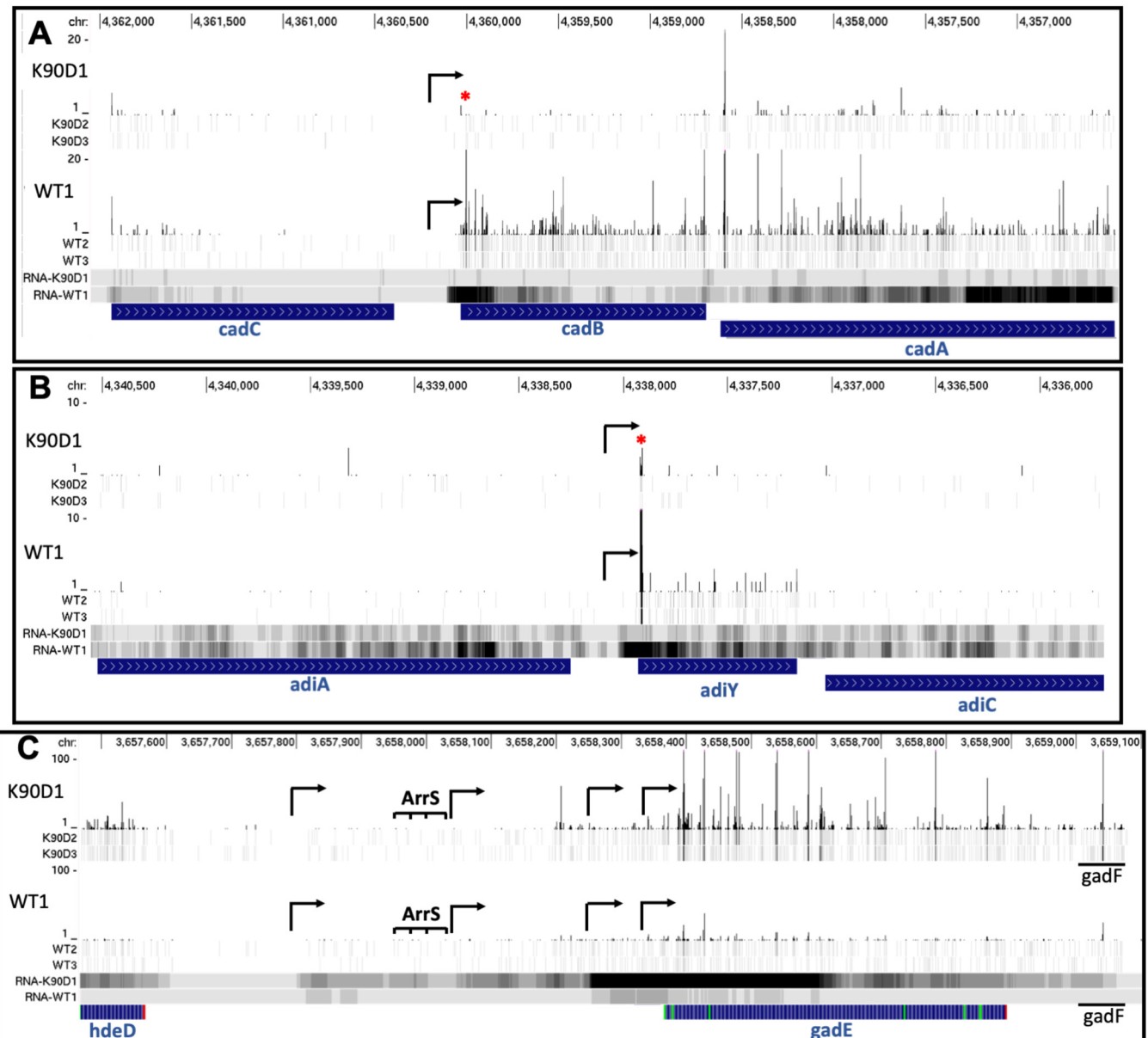

**FIG 4** RPFs and mRNA coverage profiles of acid resistance-related genes in the uL22(K90D) mutant strain. Individual read density of RPFs (top panels) and mRNA levels (bottom panels) of uL22(K90D) and uL22(WT) samples are shown. The figure was obtained using GWIPS-*viz*. Each panel is auto scaled for each gene and by group. RPFs and mRNA read count units are arbitrary. Chromosomal (Chr) positions of each gene are shown above. Reduction of RPFs signatures at the beginning of genes are indicated with red asterisks. Black arrows indicate the position of transcription initiation for genes of interest. Blue bars indicate open reading frames of genes of interest (A) *cadC-cadBA*, (B) *adiY*, and (C) *gadE*. Green lines mark methionine codon positions. Red lines indicate stop codon positions.

levels (Fig. 3C, Table 1). The *asnA* gene encodes one of the two asparagine synthetases in *E. coli* which convert aspartate into asparagine, consuming ammonia and increasing inorganic phosphate and protons as by-products (27). Additionally, expression of the *tdcA* and *tdcB* genes of the *tdcABCDEFG* operon involved in L-threonine and L-serine catabolism were significantly reduced (Table 1). Activity of this operon generates ammonia, pyruvate, and propionyl-CoA as by-products during anaerobiosis (28). All of these genes and operons seem to be affected primary via reduction of their mRNA levels.

　　Operons involved in the catabolism of the carbon sources galactarate, glycolate, fumarate, and aspartate showed low expression levels in our uL22(K90D) strain (Fig. 3A and C, Table 1). Additionally, the transcriptional regulator *cdaR* gene (29) that controls expression of the galactarate operons exhibited low expression levels as well (Table 1).

**TABLE 2** Fold increase[a] in mRNA abundance, ribosome occupancy, and translation efficiency for the expression of acid resistance genes affected in the uL22(K90D) strain

| Operons/genes | mRNA levels | RPF | RPF/mRNA levels |
|---|---|---|---|
| *mdtEF*[b] | Multidrug efflux pump | | |
| *mdtE* | 1.7 ± 0.1 | **4.7 ± 0.8**[c] | **2.8 ± 0.2** |
| *mdtF* | 1.9 ± 0.1 | **4.5 ± 0.7** | **2.3 ± 0.2** |
| *gadABC* | Glutamate decarboxylation | | |
| *gadA* | 14.0 ± 2.5 | 17.3 ± 1.2 | 1.2 ± 0.1 |
| *gadB* | 14.2 ± 2.5 | 12.5 ± 1.2 | 0.9 ± 0.1 |
| *gadC* | 10.4 ± 1.2 | 11.0 ± 1.5 | 1.1 ± 0.1 |
| *hdeAB* | Periplasmic chaperones | | |
| *hdeA* | 8.9 ± 1.4 | 8.2± 0.4 | 0.9 ± 0.1 |
| *hdeB* | 9.9 ± 1.4 | 8.6 ± 1.1 | 0.9 ± 0.1 |
| | Acid resistance membrane proteins | | |
| *yhiM* | 1.6 ± 0.3 | **9.9 ± 2.8** | **6.0 ± 0.6** |
| *hdeD* | 6.8 ± 0.9 | 7.0 ± 0.5 | 1.0 ± 0.1 |
| | Acid response transcriptional regulator | | |
| *gadE* | 8.4 ± 0.8 | 6.7 ± 0.6 | 0.8 ± 0.1 |
| *astCADBE* | Arginine →glutamate | | |
| *astC* | 5.1 ± 1.7 | 6.4 ± 3.3 | 1.3 ± 0.2 |
| *astA* | 4.9 ± 2.1 | 6.9 ± 2.7 | 1.4 ± 0.2 |
| *astD* | 4.3 ± 1.9 | 5.1 ± 3.4 | 1.3 ± 0.1 |
| *astB* | 2.6 ± 0.6 | 3.6 ± 1.4 | 1.5 ± 0.2 |
| *astE* | 2.2 ± 0.4 | **6.3 ± 5.3** | **3.3 ± 2.1** |
| | Arginine metabolism | | |
| *argC* | 1.1 ± 0.1 | **3.0 ± 0.7** | **2.7 ± 0.6** |

[a]Standard deviation calculations can be found in the supplemental file.
[b]The underline terms indicate the name of the operon which gene's values are shown in the tables.
[c]Bold numbers indicate marked reduction of translation efficiency.

However, the regulator of the glycolate degradation operon, *glcC*, did not show any significant changes in either mRNA levels or translation efficiency (see supplemental file). Interestingly, as observed for the *cadBA* operon, the first genes of many of these affected operons, *glcD*, *garP*, and *dcuB*, exhibited significant reductions in their translation efficiency in the uL22(K90D) mutant strain (Fig. 3B, Table 1). The low translation efficiency of these genes could decouple transcription from translation promoting the reduction of expression of the downstream genes of each operon.

**Genes with increased expression in the uL22(K90D) mutant strain.** Overall, we observed increased expression of genes related to acid resistance and cell signaling in the uL22(K90D) strain. Interestingly, many of the genes related to acid resistance can be found in a chromosomal cluster collectively known as the acid fitness island (21). Within this genomic region, we observed high expressional activity of the *arrS-gadE-gadF-mdtEF* loci (30). Both the *gadE* gene, which encodes a transcriptional regulator, and the *mdtEF* operon, which encodes a multidrug efflux transporter, showed an increase in both mRNA abundance and RPFs values in the uL22(K90D) strain compared with the wild-type strain (Fig. 3C, 4C, Table 2), indicating high protein expression of these genes. The transcriptional regulator GadE and the regulatory ncRNAs expressed in these loci, ArrS and GadF, control the expression of genes involved in conferring cellular resistance to acid induced stress (30, 31). These include key genes of the glutamate-dependent acid resistance system: the glutamate decarboxylase isozymes *gadA/gadB* and the antiporter *gadC*, which as expected showed high expression levels in the uL22(K90D) mutant strain comparing with the wild-type strain (Fig. 3C and Table 2). In agreement with these last observations, we observed high glutamate decarboxylase (GAD) activity in the uL22(K90D) strain compared with its parental wild-type strain (Fig. 5A, upper panel). Importantly, high GAD activity was also detected in the double mutant uL4(R61D)/uL22(K90D), but not the single mutant uL4(R61D) (Fig. S3), which

**TABLE 3** Fold change[a] in mRNA abundance, ribosome occupancy, and translation efficiency for the expression of biofilm formation related genes affected in the uL22(K90D) strain

| Operons/genes | mRNA levels | RPF | RPF/mRNA levels |
|---|---|---|---|
| *lsrRK*[c] | AI-2 internalization and metabolism | | |
| *lsrR* | 3.7 ± 0.2 | 5.9 ± 1.0 | 1.6 ± 0.2 |
| *lsrK* | 2.2 ± 0.2 | **5.7 ± 0.3** | **2.6 ± 0.1**[b] |
| *lsrACDBFG* | | | |
| *lsrC* | 1.6 ± 0.2 | **4.6 ± 1.8** | **3.0 ± 1.1** |
| *lsrD* | 1.7 ± 0.2 | **3.8 ± 0.7** | **2.2 ± 0.4** |
| *lsrB* | 5.0 ± 1.8 | **12.0 ± 4.8** | **2.3 ± 0.2** |
| *lsrF* | 4.9 ± 0.8 | 5.8 ± 1.0 | 1.2 ± 0.2 |
| *lsrG* | 4.0 ± 0.2 | 5.7 ± 0.9 | 1.4 ± 0.2 |
| | Biofilm-acid response two component connector | | |
| *ariR* | 3.7 ± 0.7 | 4.7 ± 0.8 | 1.3 ± 0.1 |
| *tnaCAB* | Indole production | | |
| *tnaC* | −2.4 ± 0.2 | −3.3 ± 0.7 | −1.4 ± 0.3 |
| *tnaA* | −2.0 ± 0.2 | −2.2 ± 0.1 | −0.9 ± 0.1 |
| *tnaA-lacZ* | −1.4 ± 0.1 | −1.6 ± 0.1 | −1.2 ± 0.1 |
| *tnaB* | 1.0 ± 0.1 | −1.1 ± 0.1 | −1.1 ± 0.1 |
| | Biofilm regulator | | |
| *bssS* | −2.1 ± 0.2 | −2.2 ± 0.2 | −1.1 ± 0.1 |

[a]Standard deviation calculations can be found in the supplemental file.
[b]Bold numbers indicate marked reduction of translation efficiency.
[c]The underline terms indicate the name of the operon which gene's values are shown in the tables.

suggests that increased expression of the GAD genes and their regulator *gadE* is a direct effect of the uL22(K90D) mutant protein expression.

We also observed an increase in the expression of additional genes within the acid fitness island, including the acid induced *hde* genes and *yhiM*, which produces a membrane protein (Fig. 3B and C, Table 2). Like the GAD genes, the expression of these genes is also regulated by GadE and confers increased resistance to acid (32). Taken together, these results reveal that in the uL22(K90D) strain there is an increase in the expression of genes related to the acid response, including membrane proteins and periplasmic chaperones, accompanied by an increase in GAD activity. In addition to the differential expression in acid related genes, the alkaline resistance *astCADBE* operon, which catabolizes arginine, producing ammonia and generating glutamate (33), exhibited increased mRNA levels as well (Table 2). Interestingly, *argC*, which is involved in the arginine biosynthetic pathway, showed increased translation efficiency in the uL22(K90D) mutant strain (Table 2), possibly contributing to the generation of glutamate by funneling arginine. Thus, in general, we observed a trend toward increasing amino acid metabolism that would modify the flux of glutamate, pyruvate, ammonia, and protons.

**Acid Resistance in the uL22(K90D) mutant strain.** Because uL22(K90D) cultures with neutral pH had high GAD activity, we sought to investigate if these changes had functional implications on the survival of our uL22(K90D) mutant strain in extremely acidic conditions. Induction of genes within the glutamate-dependent acid resistance system increases the survivability of cells exposed to acidic conditions around pH 2.0 to 2.5 (34). We reasoned that the increased GAD activity would result in a higher survival of uL22(K90D) cells at pH of 2.5 compared with wild-type uL22 cells. We challenged wild-type and uL22(K90D) cultures grown for 4 h in neutral LB media (before entering the transition into the stationary phase) to survive a shift to acidic media (pH 2.5) for 1 h. Following exposure to acidic media, we calculated the CFU per each tested culture (see Materials and Methods) and the average results can be seen in Fig. 5B. Wild-type cultures generated 2-fold more CFU than the uL22(K90D) cultures prior to the acid challenge (Fig. 5B, upper panels). We also observed that the uL22(K90D) CFU were smaller in size compared with the wild-type CFU (Fig. 5B, upper panels). After the acid challenge, we recovered approximately 0.001% CFU/mL from the wild-type

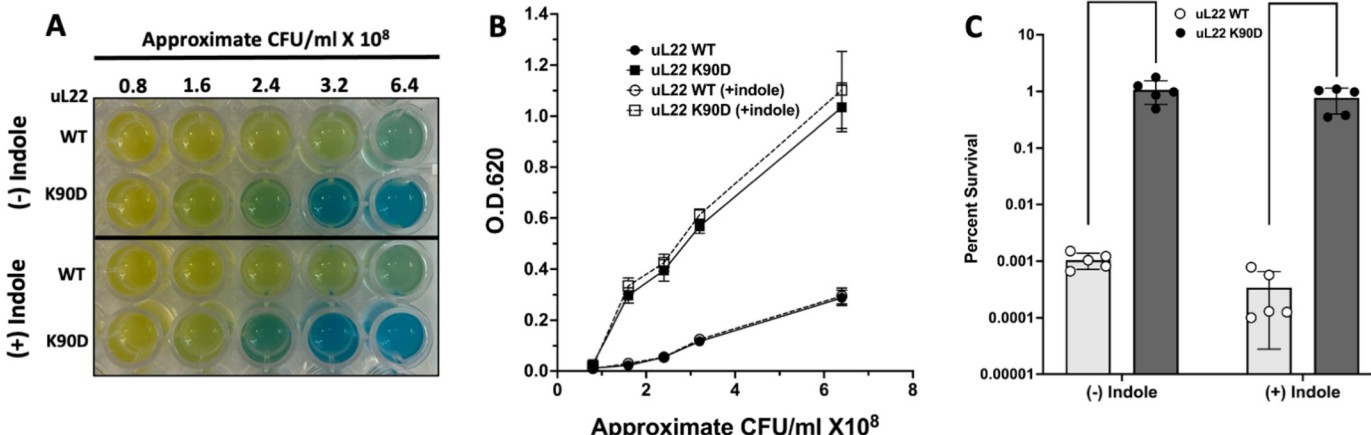

**FIG 5** Glutamate decarboxylase (GAD) activity and survival of wild type uL22 and uL22(K90D) strains in acidic conditions. (A) Representative picture of our GAD activity assays. GAD activity was determined in uL22 (WT) and uL22(K90D) cultures, with and without 0.6 mM indole following 4 h of growth in LB-pH 7 media as indicated in Materials and Methods. (B) Plot representation of the GAD activity determined by light absorption at $OD_{620}$ versus number of CFU/mL used in each reaction. Error bars represent standard deviation of $n = 4$ independent experiments. (C) uL22 (WT) and uL22(K90D) cultures, grown in LB-pH 7 media with and without 0.6 mM indole, were subjected to acidification in LB-pH 2.5 for 1 h. The CFU/mL before and after acid treatment were determined through aerobic plate counts (see Materials and Methods). Acid challenge data were expressed as percent survival obtained by dividing the CFU/mL values after acid treatment by the CFU/mL values obtained before treatment. Error bars represent standard deviation of the indicated number of independent experiments. *** $P = 0.0074$, and $P = 0.0098$, without (−) and with (+) addition of indole, respectively (Student's $t$ test).

cultures, whereas approximately 1% CFU/mL were recovered from the uL22(K90D) mutant strain (Fig. 5B, lower panel; Fig. 5C). These results indicate that the uL22(K90D) cells have a higher percentage of survival compared with the wild-type cells under exposure to extremely acidic conditions. Therefore, it is plausible that the increased GAD activity observed in the uL22(K90D) in neutral media gives these cells more opportunities to survive abrupt changes in pH compared with their parental wild-type strains.

Indole supplementation to cultures has been shown to repress expression of *gadA*, *gadB*, and *gadC* (11, 12), as well as *gadE* (11, 35), and reduce GAD activity (36). These effects impact the cell's ability to survive extreme pH (≤ 3) conditions (11, 12, 37). Thus, we hypothesized that reduced indole synthesis in uL22(K90D) cultures could lead to increased expression of these acid resistance genes. However, despite the addition of indole to wild-type and uL22(K90D) cultures, there was no discernible difference in the GAD activity between cultures with and without indole supplementation (Fig. 5A; upper panel versus lower panel). Accordingly, the supplement of indole to cultures did not increase the proportion of CFU after the acid challenge in neither the wild-type nor uL22(K90D) cultures (Fig. 5C). Overall, these data indicate that the increased expression of genes related with the glutamate-dependent acid resistance system in uL22(K90D) cultures is not due to the deficiency in indole production in this strain. As previously discussed, in the uL22(K90D) mutant strain, the *cadBA* operon displayed reductions in both mRNA abundance and RPFs, with *cadB* having one of the most significant reductions in translation efficiency (Table 1). Because the *cadBA* operon encodes the elements necessary for lysine-dependent acid resistance, we suspected that the increased expression in genes associated with the glutamate-dependent acid resistance system in uL22(K90D) may be a compensatory effect of low *cadBA* expression. For these reasons, we determined if isogenic *cadB* and *cadA* mutants had increased GAD activity. However, a *cadB* mutant strain displayed comparable GAD activity to its parental wild-type strain, while GAD activity in a *cadA* mutant was reduced compared with wild type (Fig. S4A). Analysis of the growth patterns of *cadB* and *cadA* mutant strains compared with wild type also confirmed that the reduction in *cadBA* expression does not contribute to the uL22(K90D) growth deficiency (Fig. S4B). Taken together, these data suggest that the decreased expression of the *cadBA* operon in uL22(K90D) cells is not responsible for the increased expression of GAD genes or the deficiencies in growth.

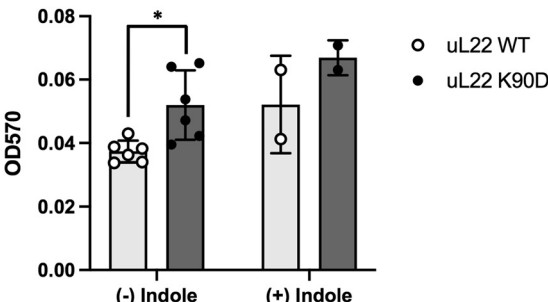

**FIG 6** Biofilm formation observed in wild-type and uL22(K90D) strains. Biofilm formation was tested in LB media in the absence ($n = 6$) or presence ($n = 2$) of 0.5 mM indole prior to 24 h of static incubation as indicated in Materials and Methods. Error bars represent standard deviation. Asterisk denotes a $P$ value of 0.010 (Student's $t$ test).

**Biofilm formation in uL22(K90D) mutant strain.** The metabolism of two important interspecies signaling molecules, indole and AI-2, appear to be affected in our mutant strain. Both indole and AI-2 are known to modulate the expression of other genes, including genes related to quorum sensing and biofilm formation (38, 39). As previously mentioned, the *tna* operon showed low expression levels (Table 3) and tryptophanase synthesis was reduced in uL22(K90D) cultures (Fig. S2) which would lead to indole deficiencies. Additionally, we observed an increase in the expression levels of the operons involved in the uptake and phosphorylation of the quorum sensing AI-2 molecule, *lsrACDBFG* and *lsrRK* (Fig. 3C, Table 3). Interestingly, the *lsrK* gene, which encodes the AI-2 kinase responsible for AI-2 internalization, showed a significant increase in its translation efficiency (Fig. 3B, Table 3). The expression of several genes whose activities function to connect indole and AI-2 signaling were also affected in the uL22(K90D) strain. This includes the gene *bssS*, whose product responds to glucose and AI-2 and functions to reduce biofilm formation and increase the uptake of indole (40), which exhibited low expression levels in the uL22(K90D) mutant strain compared with wild type (Table 3). On the other hand, the *ariR* gene, which encodes a biofilm regulator linked to indole, AI-2, and acid resistance (41), showed a significant increase in its expression levels in the uL22(K90D) mutant (Table 3). We investigated whether these changes in expression led to alterations in the formation of biofilms in cultures of the uL22(K90D) strain. Results using a standard biofilm crystal violet assay (see Materials and Methods) revealed that uL22(K90D) cells form nearly 40% more adhered biofilm cells than the wild-type cells (Fig. 6). Because indole has been implicated as a biofilm inhibitor in most *E. coli* strains (11), we were interested to see if the increased biofilm formation observed for the uL22(K90D) cultures was due to its defect in indole synthesis. If this is the case, we would expect that the addition of indole would decrease biofilm formation of uL22(K90D) cultures to levels which were observed in wild-type cultures. However, we observed that following the addition of indole to cultures, biofilm formation increased for both the wild-type and uL22(K90D) cultures (Fig. 6); the presence of external indole did not change the differences in the generation of biofilms observed between the uL22(K90D) and wild-type cultures (Fig. 6). Supporting this observation, we observed negligible differences in the formation of biofilms between a non-indole producing strain, which lacks *tnaA* expression, and its isogenic indole producing strain (Fig. S5). These results indicate that the increase in biofilm formation observed for the uL22(K90D) cells is not due to its reduction of indole synthesis and instead is perhaps dependent on the high expression of the *lsr* operons.

## DISCUSSION

In this work, we observed that a single mutational change at a non-conserved residue position of the extended loop of uL22 ribosomal protein significantly impacts cellular growth and gene expression. This mutational change substituted an acidic amino acid in a position where only basic or neutral amino acids residues are observed

among eubacteria (Fig. 1B). Cells expressing ribosomes with the uL22(K90D) mutant protein displayed growth deficiencies in the transition from the exponential phase to stationary phase when grown in rich media with a neutral pH (Fig. 2). This mutation also reduced sensitivity to erythromycin, azithromycin, and telithromycin, antibiotics that target the exit tunnel, but did not affect sensitivity for antibiotics that target other regions of the ribosome or additional cellular targets (Table S3). Changes in the expression of several genes were observed of cultures of strains expressing uL22(K90D) mutant proteins with respect cultures of strains expressing uL22 wild-type proteins. Our analyses revealed a comparable number of genes showing either reductions or increases in their expression (Fig. 3A). Genes such as *cadB* and *adiY*, both related to acid response, were among the most significantly translationally downregulated (Fig. 3B, Table 1). The mRNA transcripts of both genes displayed low ribosome occupancy at their start codons, indicating reduced initial events of translation (Fig. 4A and B). The expression of other genes related to the catabolism of alternative carbon sources (galactarate, glycolate, fumarate, and aspartate) and amino acid catabolism (*tnaA*, *asnA*, and *tdcAB*) also had reduced expression (Fig. 3, Tables 1 and 3). On the other hand, increased expression was observed for genes associated with the acid and alkaline response (Fig. 3, Table 2), the LsrR regulon involved in the uptake and phosphorylation of AI-2, and the *ariR* gene that controls both biofilm formation and the acid response (Fig. 3, Table 3). These upregulated functions appear to produce an enhancement in glutamate-dependent acid resistance (Fig. 5) and in the production of biofilms (Fig. 6). In summary, our data indicate that the K90D change in the extended loop of uL22 protein reduces the expression of particular genes, which in turn, may induce compensatory increases in the expression of genes with related activities, leading to alterations in cellular behaviors.

**Gene expression addressed by mutational changes at the uL22 protein.** Comparing our results with previous analyses of bacteria harboring mutational changes in the extended loop of uL22 protein, we found a considerable number of distinctions and commonalities. In the case of the uL22(Δ82-84) mutational change, the regulation of the *secMA* operon, whose function is important for protein translocation through the cellular membrane (42), is affected (16). However, in the case of the uL22(K90D) mutational change, we did not observe any significant changes in the expression of the *secMA* operon (see supplemental file). Both uL22(Δ82-84) and uL22(K90D) mutations produced a reduction in *tnaA* mRNA abundance, albeit the 2.5-fold reduction (Table 3) in our uL22(K90D) mutant was far more modest than the 67-fold reduction in the uL22 (Δ82-84) (16). Additionally, we did not observe changes of the major genes discussed by Yap and Bernstein for the uL22(Δ82-84) (16). These observations led us to conclude that distinctive mutational changes in the extended loop of uL22 protein generate various outcomes regarding the global expression of genes and suggests that the variability in the nature of the tunnel could have distinct impact regarding the gene being translated, as previously suggested (6).

**Mechanism of inhibition of gene expression produced by uL22(K90D).** We asked what the role of the uL22(K90D) mutant change was on the observed changes in the expression of genes. We observed an increase in the expression of the ribosome hibernating factors Rmf and SrA (see supplemental file) and a reduction in the expression of the Rmf-competitor, a hibernation factor RaiA (aka YfiA) (see supplemental file), which induces the formation of inactive ribosomes (43). We consider that under these conditions mutant ribosomes are being sequestered, which could reduce the translation initiation of genes most dependent on ribosome concentrations, including *cadB*, *adiY*, and *tnaCAB*, as well genes in the front of the *glc*, *gar*, and *dcuB-fumB* operons, among others (Fig. 7). What drives the sequestering of the mutant ribosomes is something to consider for future investigations. However, we suspect that the K90D replacement on the uL22 ribosomal protein may affect ribosome biogenesis or stabilization. Alternatively, the K90D change located at the exit tunnel constriction region could induce peptidyl-tRNA drop off at the beginning of open reading frames as seen for macrolide antibiotics that interact at the same region of the exit tunnel (44). The

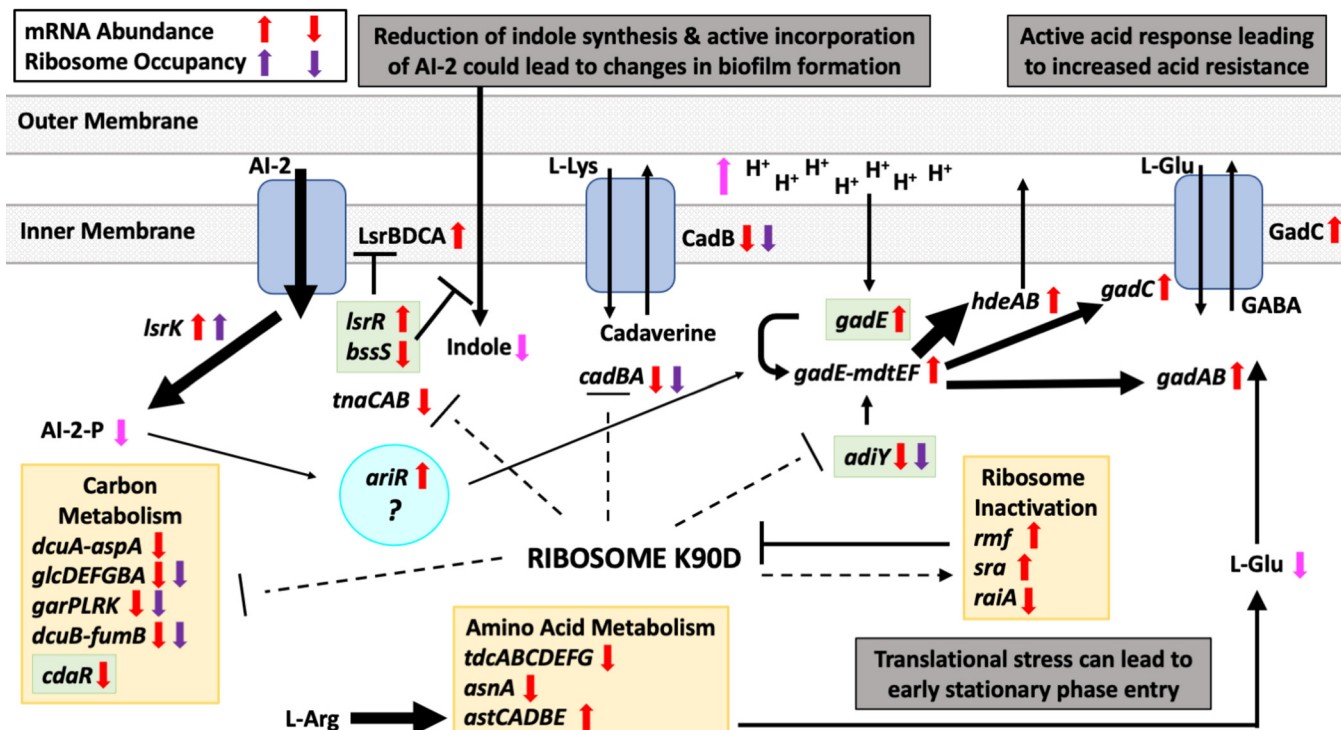

**FIG 7** Model of gene regulation observed in the uL22(K90D) mutant strain. An overview of the interconnections between genes and substrates affected in the uL22(K90D) strain. Dotted lines mark genes showing reduced translation efficiency due to the activity of ribosomes containing uL22(K90D) mutant proteins. The gray mesh boxes represent the inner cellular membrane of *E. coli*. Blue boxes represent membrane channels and transporters and arrows indicate movement of molecules across the channels and transporters. Green boxes enclose regulatory genes. Cyan circle encloses a possible key gene that controls the phenotypes observed in the uL22(K90D) mutant strain. Black arrows indicate movement of metabolites, chemical reactions, or proteins interactions with targeted genes; the thickness of the arrow is proportional to the activity of the event. Magenta arrows represent the level of accumulation expected for certain metabolites.

premature release of the ribosome on these sequences would reduce the number of ribosomes moving on their mRNAs promoting their degradation. Finally, translation of codon sequences prone to induce ribosome arrest/translational pauses such as rare codons, e.g., arginine AGA codon (45), or proline codons (46–48) could also induce the sequestering of ribosomes affecting gene expression. We observed that certain rare codons, such as AGG, AGA, and AUA, were occupied more often by mutant ribosomes. When considering genes that show a translation reduction, only *adiY* contains two rare AGG codons at the beginning of the gene. In the case of the *cadBA* operon, the transcriptional regulator *cadC* gene is known to contain proline codons in tandem (47), which could affect its expression. Our analysis did not show increases in the ribosome occupancy of such *cadC* proline sequences, neither changes in the expression of EF-P (see supplementary file) that releases the translation arrest at proline sequences (47). Despite this observation, we cannot rule out the possibility that the translational pause in the *cadC* gene is not affected. Additional *in vitro* experiments may be needed to determine if such sequences are translationally affected by the mutant ribosome.

**Acquisition of biofilm formation and acid resistance.** Our data show alterations in the expression of several genes that are controlled by an intricate combination of transcriptional factors. At this moment, we cannot discern which factor(s) are involved in producing these changes and which are consequently affected. The uL22(K90D) strain has an increased capacity to generate biofilms (Fig. 6). The two signaling molecules, indole and AI-2, have been implicated in the formation of biofilms in *E. coli* (11, 49) and seem to play opposing roles in *E. coli*. Indole is most commonly associated with inhibition of biofilm formation (11) while AI-2 stimulates biofilm formation (49). Also, indole has been implicated in cellular repulsion (11) while AI-2 is involved in cellular attraction (49). Finally, indole is secreted by cells upon transition into the stationary

phase, which corresponds to the timing in which AI-2 uptake by cells occurs (50, 51). Production of indole is solely controlled by the *tna* operon, whose expression is down-regulated through mRNA levels in uL22(K90D) cells (Table 3). Consistent with changes in the expression of *tnaA*, we observed a reduction in the synthesis of tryptophanase, the expression of the *tnaA-lacZ* reporter gene, and indole formation (Fig. S2). In our experiments, the addition of indole to both wild-type uL22 and uL22(K90D) cultures increased the formation of biofilms (Fig. 6). Additionally, our data also show reduced expression of the *bssS* gene (Table 3) which is known to induce the incorporation of indole into the cell (40). Together, these changes suggest a reduction of indole within the uL22(K90D) cell (Fig. 7), which would lead to an increase in the formation of bio-films. On the other hand, genes related to AI-2 uptake and phosphorylation were highly upregulated in the uL22(K90D) strain (Table 3), indicating a reduction of AI-2 extracellular activity. We suspect that incorporation and phosphorylation of AI-2, together with reduction of internal indole, may explain the increase in biofilm formation observed in the uL22(K90D) strain. We also observed an increase in the expression of the *ariR* gene (Table 3), whose expression regulates the formation of biofilms and also increases the expression of the *gad* operon (Fig. 7) (41). Thus, we suspect that the *ariR* gene could be a key regulator of both biofilm and acid response/resistance in our mutant strain.

In conclusion, our results indicate that a single amino acid change in the extended loop of the uL22 protein can have profound impact in the expression of genes related to the acid resistance response, metabolism, cellular indole concentrations, and the incorporation and modification of AI-2. These changes are accompanied by a defi-ciency in cell growth, an increased resistance to extreme acidic conditions, and an enhancement in biofilm formation. Ongoing and future work include determining the molecular mechanism by which the uL22(K90D) mutant is modulating changes in gene expression and translation of the discussed genes.

## MATERIALS AND METHODS

**Molecular modeling and Weblogo creation.** The structure of the *E. coli* 50S ribosomal subunit con-taining a regulatory nascent peptide (PDB 4UY8) was modeled using PyMOL (The PyMOL Molecular Graphics System, Version 2.0 Schrödinger, LLC). Weblogos were generated as previously described (52). Extended loop sequences for uL4 and uL22 homologous proteins were aligned from 231 species across 16 bacterial phyla and for 56 species across 18 eukaryotic phyla. Archaea were excluded from the align-ments. Prokaryotic (eubacteria) phyla with less than 10 species were not included in these figures and no partial, multispecies, or consensus sequences were used.

**Bacterial strains, plasmids, and growth media.** A full list of bacterial strains and plasmids used in this work can be found in Table S1. LB (5 g/L sodium chloride, 10 g/L tryptone, 5 g/L yeast extract), and LB agar (15 g/L) were used to maintain and test the studied *E. coli* strains. Acidic media was prepared by the addition of hydrochloride acid to LB media until the desired pH was reached. Adjusted pH media was then sterilized via filtration through a 0.2 $\mu$m Nalgene syringe filter (Thermo Scientific). Antibiotic stock solutions were prepared by dissolving each antibiotic in the appropriate solvent and stored at −20°C, with dilutions prepared in sterile deionized water. For maintenance and testing, the following concentrations of antibiotics were used: 10 $\mu$g/mL tetracycline (Tet) and 50 $\mu$g/mL kanamycin (Km).

**Generation of uL4 and uL22 mutant strains.** *E. coli* K-12 strain SVS1144, which carries in its chro-mosome an inserted lambda phage with a *tnaC-tnaA-lacZ* reporter gene, was used to generate uL4 and uL22 mutant strains. Initially, site-directed mutagenesis was performed targeting desired codon posi-tions of the *rplD* and *rplV* genes on the Tet-resistant pS10 plasmid (53) using the QuikChange Lightning mutagenesis kit (Agilent Technologies) and the primers indicated in Table S2. Mutated plasmids were Sanger-sequenced (Eurofins) to confirm the desired mutation. Wild-type and mutant pS10 plasmids were transformed into SVS1144. To eliminate the expression of uL4 and uL22 wild-type proteins from the chromosome of the SVS1144 strains carrying the pS10 plasmids, we transduced a Km resistance cas-sette from the SM1110 strain using P1vir bacteriophages. The SM1110 strain contains a Km resistance cassette inserted at the transcriptional promoting region of the chromosomal s10 operon, which elimi-nates its expression (53). Resulting transductants were growth-tested on LB media plates with Km and Tet to confirm the chromosomal insert of the Km^R cassette and the presence of the pS10 plasmid var-iants, respectively. Thereafter, transduced strains were grown for further experiments without Km and Tet antibiotics as suggested previously (53). Plasmids must be retained because of the need for the expression of the s10 operon, which is eliminated from the chromosome of these bacteria.

**Growth curves.** Cellular growth was measured using a Klett photoelectric colorimeter, Manostat (Cat #76-500-000). This instrument measures the intensity of scattered light, thus allowing for compari-son of the density of cells suspended in liquid media. Cultures were grown in a nephelo culture flask in 10 mL LB media. Prior to the start of each experiment, the $OD_{600}$ of overnight cultures was measured

and cultures were normalized according to cell density. Cell growth was monitored and expressed as Klett units as a function of time. Doubling times were calculated using the following formula:

$$\text{Doubling times} = [90-100 \text{ min} \times \ln 2]/[\ln(\text{Klett values at 240 min/Klett values at } 140-150 \text{ min})]$$

Averages of doubling times and standard deviations (SD) were calculated for 16 to 21 growth curves of each studied strains. Specific calculations are shown in the supplemental file.

**Ribosome profiling.** Cells were subcultured in 100 mL of LB media to an approximate absorbance of 0.6 $OD_{600}$ (corresponding to 120 to 160 Klett units). 100 $\mu$g/mL of chloramphenicol (Cm) was then added and cultures were left to grow for 3 more min. Cells were harvested via centrifugation (4,500 × $g$ for 10 min at 4°C) with a Sorvall LYNX 6000 Superspeed Centrifuge. Cell pellets were resuspended in 5 mL 1X Polysome Buffer (20 mM Tris-HCl pH 7.4, 100 mM sodium chloride, 5 mM magnesium chloride) supplemented with 100 $\mu$g/mL Cm and centrifuged again (3,220 × $g$ for 10 min at 4°C) using an Eppendorf 5804 benchtop centrifuge. The liquid was then decanted, and pellets were flash frozen using liquid nitrogen. RNA sample preparations were performed as described previously (48) using the ARTseq and Epicentre's Ribo-Zero kits (Illumina). Briefly, samples were lysed using the kit provided polysome buffer and polysomes were converted to 70S monosomes by micrococcal nuclease digestion. The 70S monosomes were separated by size exclusion chromatography and RNA was then extracted from the monosome fraction. rRNA was depleted by using the Ribo-Zero kit and PAGE purification of the ribosome protected fragments (RPFs) between 15 to 50 nt was followed by cDNA synthesis. All Ribo-seq library preparation and sequencing were performed by TB-SEQ, Inc. on an Illumina NovaSeq6000 sequencer. Both raw and trimmed (of adaptor sequences) RNA-seq and Ribo-seq libraries were generated and quality control data was provided for each sample in both library sets.

**Ribosome profiling analysis.** First, all trimmed fastQ sequences were confirmed to correspond to the appropriate wild-type or uL22(K90D) cell samples using the Grep command in Linux. The trimmed sequences were then processed using two different pipelines. The RiboGalaxy and RiboToolkit web tools were used to visualize and quantify the RNA-seq and ribosome profiling sequence data, respectively (54, 55). Each tool processed the same raw data, with the qualitative read count visualization from RiboGalaxy matching the quantitative outputs of RiboToolkit. Adapter trimmed fastq sequences were first uploaded to RiboGalaxy and the read quality was verified using FastQC (56). All sequence data displayed high-quality reads up to 40 bp (data not shown). Once quality was confirmed, rRNA was removed from the samples using Bowtie (57) and then the remaining reads were aligned to the provided reference genome. The default settings were used as well as the *E. coli* str. K-12 substr. MG1655: eschColi_K12 reference index and genome. The aligned file output was converted from SAM to BAM format and then a ribosome profile (GWIPS-viz Mapping tool) was generated using an offset from the 3′-end with default settings corresponding to an A-site (of 12) offset and the eschColi_K12 reference genome (GenBank # GCA_000005845). The resulting bigwig files were visualized through GWIPS-viz (58).

RiboToolkit is an online resource for analyzing RNA- and ribo-seq data. Manipulating the data for input was done on an Ubuntu 20.04.1 LTS server accessed remotely using PuTTY with all necessary packages, libraries, and dependencies downloaded directly to the server. Genome assembly ASM584v2 was downloaded from https://bacteria.ensembl.org and used as a reference genome for alignment (59). Inputs for RiboToolkit were produced by following the suggested preparations (sequences were already trimmed and cleaned upon receipt). The utility to create the collapsed FASTA files was downloaded and implemented through Perl to generate the inputs for the single case ribo-seq analysis. The featureCounts utility from the subread package was used to create a gene count matrix of the RNA counts of all samples to generate the input for the group case study (60, 61). The BAM files for input into featureCounts were generated using Bowtie (1.2.3) default settings for read count alignments and SAMtools (1.10) for SAM to BAM format conversion (62). *E. coli* K-12 was used as the selected species for all analysis/tools in RiboToolkit. Single case analysis of each Ribo-seq sample was performed using the default settings except for RPF length, which was broadened to 20 (shortest) and 40 (longest). Next, group analysis was performed using all the previous single case results combined with the RNA-seq data to yield information about differential translation between groups using default settings. The averages of the read counts for both mRNA and RPFs for each sample group (wild type or K90D) were determined. Fold change in mRNA abundance and ribosome occupancy were calculated as the ratio mRNA of uL22(K90D)/mRNA of wild type, and RPFs of uL22(K90D)/RPFs of wild-type average read counts for each gene, respectively. These ratio values were also used to determine the adjusted *P*-values (corrects for false discovery rate). Translation efficiency was calculated as the ratio RPFs/mRNA read counts per each gene. Fold change in translation efficiency was calculated as the ratio translation efficiency of uL2(K90D)/translation efficiency of wild-type values (55). Standard deviations of the ratios (SD ratio) shown in Tables 1 to 3 were calculated using the following equation:

$$\text{SD ratio} = (\text{values of K90D/values of wild type}) \times \ddot{O}(\text{SD K90D/values K90D})^2$$
$$+ (\text{SD wild type/values wild type})^2 - (2 \times \text{Covariance})/(\text{values K90D} \times \text{values wild type})$$

$$\text{Covariance} = (\text{Correlation of values K90D and wild type}) \times (\text{SD K90D}) \times (\text{SD wild type})$$

**Glutamate decarboxylase assays.** The activity of glutamate decarboxylase was determined using a previously described assay with some modifications (63). Subcultures, prepared from normalized

dilutions of overnight cultures, were grown at 37°C in a shaking water bath. After 4 h of incubation, the cultures were removed from the water bath and spun down (4,000 × *g* for 20 min at 4°C) with an Eppendorf 5810R Centrifuge. The resulting pellets were resuspended in one mL of 0.85% sodium chloride. The cell suspension was then centrifuged at 14,000 rpm for 5 min in an Eppendorf miniSpin centrifuge. The pellet was resuspended in 0.85% sodium chloride for a final volume of 500 $\mu$L. The cellular suspensions were then normalized and diluted so that they had approximately equal $OD_{600}$ measurements. Various aliquots of the normalized cell suspensions were added to 500 $\mu$L of GAD reagent (0.05 g/L bromocresol green, 0.9 g/L potassium l-glutamate, 90 g/L sodium chloride, 3 mL/L Triton X-100, 4 mM hydrochloric acid) and incubated for 1 h at 37°C. An aliquot of the reaction was pipetted into a 96-well plate for visualization and comparison. Light absorption at $OD_{620}$ for each well was determined using a BIO-TEK PowerWaveHT spectrophotometer microreader.

**Acid challenge assays.** Acid challenges were performed as previously described with some modifications (34). Subcultures, prepared from normalized dilutions of overnight cultures, were grown at 37°C in a shaking water bath for 4 h. Thereafter, 200 $\mu$L of each culture was added to 1.8 mL of either LB-pH 7 or LB-pH 2.5. The cellular suspensions in LB-pH 2.5 were returned to the 37°C water bath and incubated for 1 h. Final cultures were serial diluted in either LB-pH 7 or LB-pH 2.5 and various volumes were plated on LB-pH 7 agar. All plates were incubated overnight at 37°C. After 16 to 18 h, the CFU on the plates were tallied and recorded. Data were expressed as percentage survival calculated using the following formula: (CFU/mL after acidification/CFU/mL before acidification) × 100.

**Biofilm assays.** Biofilm formation assays were performed as previously described (64). One mL of normalized dilutions of overnight cultures was added to 14 mL polypropylene round-bottom tubes (FALCON) and incubated at 37°C in a static upright position for 24 h. Following incubation, planktonic cells were removed from the tubes by a series of three washes with deionized water. After tubes were air-dried, 1.5 mL of 0.1% Gram crystal violet (Remel) was added and used to stain adherent cells for 15 min. The crystal violet solution was removed and the tubes were washed once more with deionized water three times and left to air dry. Once dry, 1.5 mL of an ethanol:acetone (80:20) solution was added to tubes as a solubilizing agent and left at room temperature for 15 min. $OD_{570}$ absorbance of the resulting solution was measured in duplicate for each tube using a BIO-TEK PowerWaveHT spectrophotometer microreader. Statistical analyses and graphical representations for growth curves, acid resistance, and biofilm formation assays were obtained using GraphPad Prism 9. Data was analyzed using two-tailed unpaired Student's *t* test with an alpha value of 0.05.

**Data availability.** Ribosome profiling data was analyzed using RiboToolkit, a free webtool available at (http://210.73.222.207/RiboToolkit/). Data was also prepared for visualization on a genome viewer using RiboGalaxy, a free webtool available at (https://galaxyproject.org/use/ribogalaxy/).

The Ribo-seq and RNA-seq data reported in this paper have been deposited in the NCBI Gene Expression Omnibus (GEO) database with accession number GSE171533.

## SUPPLEMENTAL MATERIAL

Supplemental material is available online only.

**SUPPLEMENTAL FILE 1**, XLSX file, 3.6 MB.

**SUPPLEMENTAL FILE 2**, PDF file, 1.9 MB.

## ACKNOWLEDGMENTS

We thank Kevin D. Young for supplying the plasmids pLP8 and pTnaA, and Sean Moore for the plasmid pS10, phage P1vir, and the cell line SM1110. L.R.C.-V. would like to thank the UAH provost office, college of science dean, and Paul Wolf chair of the biology department for their financial support used in the preparation of the ribosome profiling assays.

The work is partially supported by NIH Grant R01GM121359 (to M.-N.F.Y.).

We declare that there is no conflict of interest.

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
