## [Reviewer comments · Microbiology Spectrum]

Microbiology Spectrum

The identity of the constriction region of the ribosomal exit tunnel is important to maintain gene expression in *Escherichia coli*

Sarah Worthan, Elizabeth Franklin, Chi Pham, M.-N. Frances Yap, and Luis Cruz-Vera

Corresponding Author(s): Luis Cruz-Vera, University of Alabama in Huntsville

Review Timeline:

Submission Date:	November 15, 2021
Editorial Decision:	December 23, 2021
Revision Received:	February 16, 2022
Accepted:	February 22, 2022

Editor: Amanda Oglesby

Reviewer(s): Disclosure of reviewer identity is with reference to reviewer comments included in decision letter(s). The following individuals involved in review of your submission have agreed to reveal their identity: Peter Adrian Lund (Reviewer #2)

Transaction Report:

DOI: <https://doi.org/10.1128/spectrum.02261-21>

December 23, 2021

Dr. Luis Rogelio Cruz-Vera
University of Alabama in Huntsville
Biological Sciences
301 Sparkman
Huntsville, AL 35899

Re: Spectrum02261-21 (The identity of the constriction region of the ribosomal exit tunnel is important to maintain gene expression in Escherichia coli)

Dear Dr. Luis Rogelio Cruz-Vera:

Thank you for submitting your manuscript to Microbiology Spectrum. Your manuscript has been reviewed by three experts in the field. While the reviews are overall positive, a few points are made that should be addressed prior to further consideration. Some of these can be addressed through text-only modifications, but others may require additional analysis and interpretation of the current data set. I will be more than happy to consider a revised manuscript that addresses these concerns.

Link Not Available

Sincerely,

Amanda Oglesby

Journals Department
Reviewer comments:

Reviewer #1 (Comments for the Author):

The MS by Worthan et al. characterizes the effects on gene expression of a mutation in ribosomal protein L22. The altered residue, K90, lies at the constriction region of the exit tunnel of the ribosome. RNAseq and ribosome profiling were used to characterize the effects on gene expression of the altered ribosomal protein. Genes involved in pH regulation and carbon and amino acid metabolism showed reduced expression, while expression of acid-induced membrane proteins and chaperones, the glutamatedecarboxylase regulon and the autoinducer-2 metabolic regulon were all increased. Some of these difference in gene expression could be correlated with altered phenotypes in the mutant.

While the data raise many unanswered questions, it is nonetheless worthwhile to have a catalogue of the genes affected by a single ribosomal change. There are multiple examples of 'ribosomopathies' in mammalian systems, where single changes in the translation system provoke profound, but nonetheless rather specific changes in gene expression. It is good that analogous changes in bacterial ribosomes are now being characterized using modern profiling methods.

The obvious question raised by the results is what is the mechanism underlying the alterations in gene expression? Does the L22 mutant translate slower, or are its effects due to an altered peptide tunnel, or some other, less obvious characteristic of the mutant ribosomes? One way of approaching this might be to see if any of the scorable phenotypes of the K90D L22 mutant (acid resistance, biofilm formation) are shared by other, better characterized ribosomal mutants. Along these lines, I note that the main author has previously published on the isolation of another L22 mutation at the same position (K90W); does the K90W also show altered acid resistance or biofilm formation?

Does the L22 K90D mutant display an altered sensitivity to any of the 50S subunit-targeting antibiotics (erythromycin and others)?

Fig 2 growth rates: Overall, growth in rich media of all strains is rather slow (~40 min doubling time for wild type), much slower than I would expect for a normal E. coli strain growing in LB at 37°C. This needs to be addressed in the text. What is the growth rate of the unmanipulated, parental strain ?

The manipulations needed to generate the L4 and L22 mutations are not well explained. It appears that the chromosomal S10 operon has been deleted in the presence of a plasmid-borne copy of the operon, expressing wild type or mutant L22 (or L4). There are now other ways of introducing mutations into chromosomal genes that are preferable to the method employed here. In any case, this manipulation should be described in more depth than a mere reference to the original Moore et al. PNAS publication.

Reviewer #2 (Comments for the Author):

This is an interesting work that has been well executed, and the write up is both thorough and easy to read.

The paper is largely observational, looking at the impact of a particular mutation which affects the ribosome exit tunnel. The changes in gene expression and ribosomal profiling show some consistency in the types of genes that are affected, and the possible consequences of this are looked at in some cases - acid resistance and biofilm formation, and the impact of indole levels on these. The RNAseq and ribosomal profiling data show good correlation and also allow the authors to drill down to genes where the impact on overall expression (and hence presumably on actual protein levels, which is checked by enzyme assay for the Gad genes) is at the level of translation.

It's noticeable that there is not a clear mechanism demonstrated for the effects that are observed. The authors makes some speculations in the discussion about this and present a possible model, but the evidence to directly support this is rather scant at this stage. The issue here I think is that at least some of the pathways they are looking at are already known to be connected in complicated ways (they refer to an example of this earlier when they cite papers showing how regulators of the different acid resistance pathways show evidence of cross-talk), and so it is very hard to disentangle the responses and point to an over-riding cause.

One possible mechanism which I think they should consider in their discussion and which I don't think they have covered is translational pausing. It has been shown by Kirsten Jung's group that the occurrence of particular combinations of proline codons can have a profound effect on the levels of some proteins due to translation pausing, including the CadC regulator protein and I believe also the EvgS sensor kinase of the Gad system. As such, it may be that they are seeing another aspect of this (it seems not impossible that the mutation they look at may induce or amplify pausing) and I think this should also be considered, even if the result is to reject this as a hypothesis that is relevant here.

A couple of specific comments:

1. The impact of indole on induction of the gad genes by mild acid shock is quite media specific (it does not occur in LB, for example) so it's not clear to me that their experiments on the impact of indole, which were done at pH 7 in LB, would have been expected to show anything.
2. In figure 5, I don't think we need to be shown the plates - they don't add to the results as the overall results are shown in 5C. So I'd delete figure 5B.

Reviewer #3 (Comments for the Author):

Attached as a separate file.

Staff Comments:

Preparing Revision Guidelines

Please return the manuscript within 60 days; if you cannot complete the modification within this time period, please contact me. If you do not wish to modify the manuscript and prefer to submit it to another journal, please notify me of your decision immediately so that the manuscript may be formally withdrawn from consideration by Microbiology Spectrum.

The manuscript by S. Worthan *et al.* describes the physiological impacts of altering semi-conserved amino acids in the exit tunnel loop regions of the uL4 and uL22 ribosomal proteins. These tunnel contact regions play important roles in interrogating nascent proteins to initiate allosteric changes that can alter protein synthesis via translational stalling or antibiotic binding. Alterations to these proteins can influence the production of many proteins, which can lead to complicated pleiotropic effects. As such, investigations into the functions and impact of these proteins continues to garner interest because they can alter a substantial portion of a cell's proteome. The authors approached their investigation using transcriptome analyses and ribosome profiling experiments. The focus was set on a culturing stage at which the growth rate of one of the mutants (uL22 K90D) began to depart from the parental strain and they were able to identify several mRNAs whose levels and/or ribosome occupancy were altered compared to the parental strain.

The report conveys important information that will be useful to the bacterial research community, but there are issues with the experimental design, data analysis, and interpretation that substantially weaken the presented material.

Major issues:

The decision to investigate the 4 h culture was based on the observation that the growth rate of the uL22 K90D culture began to slow down compared to the parental strain. However, the conclusions drawn from the transcript/profiling data are presented as though those changes were responsible for the growth rate change, rather than a consequence of it. Without a comparison to the transcript/profile levels from cultures before or after that 4 h transition, there is no way to determine correlation. For example, perhaps the expression or translation of some of those messages was even more distorted prior to the slow down and the cells were in the process of correcting them once the '4 hour shock' occurred.

The parental strain (SVS1144) that was used to compare the uL4 and uL22 mutants was not wild-type. It is reported to contain a large deletion from *lac* to *argF* (~75 kb) that includes many genes involved in metabolic regulation. Moreover, this strain contains a *tnaC-tnaA'-lacZ* fusion integrated by phage Lambda. How were these transcripts and RPF sequence reads distinguished from those originating *tnaCAB* operon? The *tnaB* mRNA and RPF did not change while those from *tnaC* and *tnaA* did. Is there an explanation for that observation? Did LacZ production change in the uL22 K90D mutant? Moreover, the parental strain used for the *cadA* and *cadB* experiment was changed to BW25110, confounding interpretations related to the uL4 and uL22 mutants.

The supplementary data that gave rise to the reported fold changes and associated standard deviations (SD) was calculated incorrectly. The authors computed a ratio of each mutant replicate to each wild-type replicate in all combinations, and then averaged those results and used those to establish SDs. The wild-type and mutant cultures were independent. The three wild-type and three mutant replicate measurements should have been averaged first. They are biological replicate measurements used to establish a value independent of the other set. Doing so also allows for a t-test between the data

sets to be more simply computed by comparing the three wild-type values to the three mutant values to identify significant changes.

An example analogy would be in the measurement of the length and weight of a cable. The three length measurements would be averaged, then the three weight measurements, then a single ratio determined that includes the variance (SD) of the two averages. For multiplication or division of numbers that contain known error (to establish fold changes, for example), a coefficient of variation (CV) should be calculated, squared, and used for computing the resulting CV of the ratio. That CV should then be converted to an SD for the final reported number.

All of this being said, the resulting numbers do not change by a lot in the case of the presented data because the CVs of the data sets were comparable (e.g., the fold change data for *glcD* is reported as 6.25 +/- 2.87, which would change to 5.85 +/- 3.05). Nonetheless, a correction of the data would be more accurate, allow proper t-tests to be computed, and also contain fewer operations on the spreadsheets.

Minor issues:

In Figure 1, can the panel be flipped such that 'upper' is on the top and 'lower' is on the bottom? (or reverse the terminology, maybe use proximal and distal relative to the PTC). Not an issue, but considering the conservation of K90 as a histidine in eukaryotes, that would also be an interesting mutant to evaluate in bacteria.

For Figure 2, The doubling time is too slow for a healthy *E. coli* in LB at 37 C, suggesting either that the parental strain was compromised or that air or some nutrient was limiting. How was the growth rate calculated? The time window is reported, but no log2 was calculated as far as I can tell. Why was the Klett data plotted by converting the Y-axis scaling instead of converting the data to log2 and fitting a line? Why were the growth rates calculated from 160-260 minutes? This region overlaps with the observed growth rate reduction by the mutant. Perhaps an analysis both pre- and post-transition would better reflect the impact of the mutation on growth rate reduction. Unfortunately, the Klett values early in the growth curves are very noisy until ~20-32 units after 200 mins of culturing, which is generally past exponential phase under these conditions.

Figure 4, the text and data are very hard to see. Can the important text and plot region sizes be increased?

Figure 5A, the legend indicates the GAD assay was performed multiple times, but there are no values, averages, or SDs reported.

Fig. 5C, the legend states that there were 3 cultures with indole and 4 without, but the plotted data shows 4 and 5 data points respectively. The survival clusters are ~100 fold different between the wild-type and mutant in each case, why is the p-value so high? Off hand, it seems you may have a 2 or 3 * P-value in each. How was the t-test performed and which data were used (the CFU or the resulting percentages)?

Fig. 6 and related results. For Figure S2, how were the identities of the indicated spots verified as TnaA and Cdd? The RNAseq data indicated that *tnaA* expression was lowered ~2 fold, but the band for TnaA protein is nearly absent compared to the parental strain in the 2D gel (yet reported as 59%). In either case, why is it asserted repeatedly that this mutant strain cannot produce indole or that there are indole deficiencies? TnaA converts imported tryptophan to indole and it requires TnaB to do so. If the TnaB levels are the same, then the indole production is expected to be the same. Your medium provided tryptophan, was it consumed?

The methods section could benefit from general editing. As examples, "hydrochloride acid" should be hydrochloric acid, "RPMs" should be RPM, and "L-glutamic acid potassium salt monohydrate" should be potassium glutamate, as it is dissolved.

Line numbering helps with commenting.

Worthan et al_ Answers to reviewer's comments

Dear Dr. Oglesby
Editor, Microbiology Spectrum

Thanks for giving us the opportunity to revise our manuscript and for yours and the reviewers' thoughtful comments to our work.

Below you would find our point-to-point responses to the questions, observations, and concerns raised by the reviewers.

We believe that we have addressed all comments and expect that the new version of the manuscript to be suitable for publication.

Thank you very much for all your help,

Best Regards

Luis Rogelio (Roger) Cruz-Vera

Reviewer #1 (Comments for the Author):

The MS by Worthan et al. characterizes the effects on gene expression of a mutation in ribosomal protein L22. The altered residue, K90, lies at the constriction region of the exit tunnel of the ribosome. RNAseq and ribosome profiling were used to characterize the effects on gene expression of the altered ribosomal protein. Genes involved in pH regulation and carbon and amino acid metabolism showed reduced expression, while expression of acid-induced membrane proteins and chaperones, the glutamatedecarboxylase regulon and the autoinducer-2 metabolic regulon were all increased. Some of these difference in gene expression could be correlated with altered phenotypes in the mutant.

While the data raise many unanswered questions, it is nonetheless worthwhile to have a catalogue of the genes affected by a single ribosomal change. There are multiple examples of 'ribosomopathies' in mammalian systems, where single changes in the translation system provoke profound, but nonetheless rather specific changes in gene expression. It is good that analogous changes in bacterial ribosomes are now being characterized using modern profiling methods.

The obvious question raised by the results is what is the mechanism underlying the alterations in gene expression? Does the L22 mutant translate slower, or are its effects due to an altered peptide tunnel, or some other, less obvious characteristic of the mutant ribosomes?

Answer: Based on our data, we do not believe that the L22 mutant ribosomes translate ALL mRNA at a slower rate than the wild type ribosomes in part because our analyses indicate that some mRNAs are translated by the mutant ribosomes at a higher rate (e.g. *yhiM*, *IsrK*, *mdtE* and *mdtF* among others, see Figure 3B). However, our data suggest that the effects of the mutant ribosome are specific to the nature of the mRNA. We also suspect that the translation rate of specific mRNA is affected by the structural alterations in the peptide tunnel as explained in our discussion (page 11, second paragraph). Indeed, additional work is required, e.g. *in vitro* cell-free translation assays, in order to definitively determine why the mutant ribosomes preferentially affect a particular subset of genes.

One way of approaching this might be to see if any of the scorable phenotypes of the K90D L22 mutant (acid resistance, biofilm formation) are shared by other, better characterized ribosomal mutants. Along these lines, I note that the main author has previously published on the isolation of another L22 mutation at the same position (K90W); does the K90W also show altered acid resistance or biofilm formation?

Answer: As discussed in our manuscript, an *E. coli* strain harboring the well characterized uL22(Δ 82-84) mutant protein does not affect the same pool of genes as our uL22(K90D)mutant, suggesting that different mutations in the uL22 protein is likely to produce different expression profiles (please see last paragraph of page 10 and first paragraph page 11). The reviewer's suggestion of using the K90W mutant is very interesting from the perspective of selectively testing the contribution of lysine residue at the 90th position. To this end, we have not introduced the K90W allele or other K90 mutations in the same genetic background. Our main intention with the current manuscript is to bring the attention to the readers that any substitutions at the 90th position could produce diverse secondary effects on the expression of genes and phenotypes in bacteria.

Worthan et al_ Answers to reviewer's comments

Does the L22 K90D mutant display an altered sensitivity to any of the 50S subunit-targeting antibiotics (erythromycin and others)?

Answer: We have not performed *in vitro* assays to evaluate the sensitivity of these ribosomes to antibiotics. However, we have tested *in vivo* the resistance of our strains bearing the mutant ribosomes to several ribosome-targeting and cell wall-targeting antibiotics. We observed that strains expressing the L22 K90D protein variant show increases in their MIC values for erythromycin, azithromycin, and telithromycin. The enhanced resistance is modest (2- to 4-fold), but significant and very reproducible (Table S3). Currently we cannot discern if the MIC changes are a product of reducing the binding affinity of these particular antibiotics for the ribosome or if they are due to increases in the expression of the *mdtE* and *mdtF* genes which produce a multidrug efflux pump, or a combination of these effects. We added our MIC values to the supplementary data, as well as an explanation of our observations in the text (page 10, lines 17-20).

Fig 2 growth rates: Overall, growth in rich media of all strains is rather slow (~40 min doubling time for wild type), much slower than I would expect for a normal *E. coli* strain growing in LB at 37°C. This needs to be addressed in the text. What is the growth rate of the unmanipulated, parental strain?

Answer: Because this question has been addressed by the other reviewers as well, we decided to re-evaluate our data using more growth curves with different time points in which there is a linearity of the curve under logarithmical values. The new calculations are presented in Figure 2A. We also added additional information on how we determined these values in Material and Methods (Page 14, lines 1-6). We observed that these strains and the parental strain (not shown) showed doubling times around 35 min. Still, it is slow, we addressed such in the text as well (legend of Figure 2).

The manipulations needed to generate the L4 and L22 mutations are not well explained. It appears that the chromosomal S10 operon has been deleted in the presence of a plasmid-borne copy of the operon, expressing wild type or mutant L22 (or L4). There are now other ways of introducing mutations into chromosomal genes that are preferable to the method employed here. In any case, this manipulation should be described in more depth than a mere reference to the original Moore et al. PNAS publication.

Answer: As suggested, we have rewritten this section in the main text to clarify the strain construction (page 13, lines 24-35). We were unsuccessful in introducing the K90D allele to the native *rpIV* locus via λ Red recombination by selecting erythromycin resistant recombinants. The lack of selection is likely due to the subtle change in erythromycin resistance of the chromosomally encoded uL22(K90D), consistent with the modest erythromycin resistance of the plasmid-borne uL22(K90D) (Table S3). CRISPR-Cas gene editing is another option, but the technology has not been established in our group and will require extensive trial and error.

Reviewer #2 (Comments for the Author):

This is an interesting work that has been well executed, and the write up is both thorough and easy to read.

The paper is largely observational, looking at the impact of a particular mutation which affects the ribosome exit tunnel. The changes in gene expression and ribosomal profiling show some consistency in the types of genes that are affected, and the possible consequences of this are looked at in some cases - acid resistance and biofilm formation, and the impact of indole levels on these. The RNAseq and ribosomal profiling data show good correlation and also allow the authors to drill down to genes where the impact on overall expression (and hence presumably on actual protein levels, which is checked by enzyme assay for the Gad genes) is at the level of translation.

It's noticeable that there is not a clear mechanism demonstrated for the effects that are observed. The authors makes some speculations in the discussion about this and present a possible model, but the evidence to directly support this is rather scant at this stage. The issue here I think is that at least some of the pathways they are looking at are already known to be connected in complicated ways (they refer to an example of this earlier when they cite papers showing how regulators of the different acid resistance pathways show evidence of cross-talk), and so it is very hard to disentangle the responses and point to an over-riding cause.

Answer: We agree with the reviewer that these pathways have intricately connected connections and that our data cannot discriminate the key factor(s) involved in the observed phenotypes. We added a couple of sentences in our discussion indicating such problematic analysis. Our intentions in our discussion are to drive the reader to connect the structure of ribosome with these pathways and suggest possible scenarios (page 12, lines 2-4).

Worthan et al_ Answers to reviewer's comments

One possible mechanism which I think they should consider in their discussion and which I don't think they have covered is translational pausing. It has been shown by Kirsten Jung's group that the occurrence of particular combinations of proline codons can have a profound effect on the levels of some proteins due to translation pausing, including the CadC regulator protein and I believe also the EvgS sensor kinase of the Gad system. As such, it may be that they are seeing another aspect of this (it seems not impossible that the mutation they look at may induce or amplify pausing) and I think this should also be considered, even if the result is to reject this as a hypothesis that is relevant here.

Answer: We agree with the reviewer. We have analyzed the translation arrest at the *cadC* gene (seen in Figure 4), but did not get a clear result if such arrest signal was affected. What we have observed, however, is that the mutant ribosomes protect specific rare codons, such as AGG and ATA, more often (shown now in Figure S4). As suggested, we added a discussion about translation arrest (page 11, starting at line 29).

A couple of specific comments:

1. The impact of indole on induction of the *gad* genes by mild acid shock is quite media specific (it does not occur in LB, for example) so it's not clear to me that their experiments on the impact of indole, which were done at pH 7 in LB, would have been expected to show anything.

Answers: We were expecting based on previous results that the addition of indole to *E. coli* cultures grown in LB would reduce the expression of the *gad* genes (Lee et. al., J Mol Biol 2007. 373:11), reducing the survival of the cells under acid challenges in LB as well (Lee et. al., J Mol Biol 2007. 373:11; Minvielle et. al, Chemistry 2013. 19:17595). We have also added more information in the text to support our case (page 8, lines 36-38).

2. In figure 5, I don't think we need to be shown the plates - they don't add to the results as the overall results are shown in 5C. So I'd delete figure 5B.

Answer: We agree with the reviewer and have deleted the Figure 5B.

Reviewer #3 (Comments for the Author):

The manuscript by S. Worthan *et al.* describes the physiological impacts of altering semi-conserved amino acids in the exit tunnel loop regions of the uL4 and uL22 ribosomal proteins. These tunnel contact regions play important roles in interrogating nascent proteins to initiate allosteric changes that can alter protein synthesis via translational stalling or antibiotic binding. Alterations to these proteins can influence the production of many proteins, which can lead to complicated pleiotropic effects. As such, investigations into the functions and impact of these proteins continues to garner interest because they can alter a substantial portion of a cell's proteome. The authors approached their investigation using transcriptome analyses and ribosome profiling experiments. The focus was set on a culturing stage at which the growth rate of one of the mutants (uL22 K90D) began to depart from the parental strain and they were able to identify several mRNAs whose levels and/or ribosome occupancy were altered compared to the parental strain.

The report conveys important information that will be useful to the bacterial research community, but there are issues with the experimental design, data analysis, and interpretation that substantially weaken the presented material.

Major issues:

The decision to investigate the 4 h culture was based on the observation that the growth rate of the uL22 K90D culture began to slow down compared to the parental strain. However, the conclusions drawn from the transcript/profiling data are presented as though those changes were responsible for the growth rate change, rather than a consequence of it. Without a comparison to the transcript/profile levels from cultures before or after that 4 h transition, there is no way to determine correlation. For example, perhaps the expression or translation of some of those messages was even more distorted prior to the slow down and the cells were in the process of correcting them once the '4 hour shock' occurred.

Answer: We understand the reviewer's concerns and we would definitely like to analyze additional points of growth to see how gene expression is changing over time. Unfortunately, ribosome profiling experiments are extremely cost prohibited and will require more time. We have re-written sections of the conclusion to eliminate any suggestion that these changes are the reason for the differences in growth (page 10, lines 20-22).

Worthan et al_ Answers to reviewer's comments

The parental strain (SVS1144) that was used to compare the uL4 and uL22 mutants was not wild-type. It is reported to contain a large deletion from *lac* to *argF* (~75 kb) that includes many genes involved in metabolic regulation.

Answer: We have changed every text that indicates the strain expressing wild type uL22 proteins are "wild type strains" by "uL22(WT)" or "wild type uL22".

Moreover, this strain contains a *tnaC-tnaA'-lacZ* fusion integrated by phage Lambda. How were these transcripts and RPF sequence reads distinguished from those originating *tnaCAB* operon?

Answer: The transcript and RPF read values for the reporter gene, *tnaA'-lacZ'* protein fusion (reported values as *lacZ* at the supplemental file) showed a similar tendency as the values reported for the *tnaA* gene (data added to Table 3). Furthermore, the activity of LacZ is observed to be reduced; we added this data to the supplementary data (Figure S2C). We have also added an explanation in the text about these observations (Page 6, lines 25-42).

The *tnaB* mRNA and RPF did not change while those from *tnaC* and *tnaA* did. Is there an explanation for that observation?

Answer: Both values, mRNA and RPF's, for the *tnaB* gene are quite low in both strains compared to the values for the *tnaC* and *tnaA* genes (please check supplemental file). It appears that the expression of *tnaB* is divorced from the very high expression of both the *tnaC* and *tnaA* genes. It is known that RNase P processed the operon at the front of the *tnaB* gene (Li and Altman, PNAS 2003. 100:13213), which decouple the stability of the *tnaA* and *tnaB* mRNAs. We have added an explanation in the text regarding these observations (Page 6, lines 25-42).

Did LacZ production change in the uL22 K90D mutant?

Answer: Yes, LacZ production is reduced in the strain expressing the K90D mutant protein. We added the experimental data measuring LacZ activity in both strains to the manuscript (supplementary data Figure S2C).

Moreover, the parental strain used for the *cadA* and *cadB* experiment was changed to BW25110, confounding interpretations related to the uL4 and uL22 mutants.

Answer: We decided to remove these results and their interpretation in order to avoid further confusion.

The supplementary data that gave rise to the reported fold changes and associated standard deviations (SD) was calculated incorrectly. The authors computed a ratio of each mutant replicate to each wild-type replicate in all combinations, and then averaged those results and used those to establish SDs. The wild-type and mutant cultures were independent. The three wild-type and three mutant replicate measurements should have been averaged first. They are biological replicate measurements used to establish a value independent of the other set. Doing so also allows for a t-test between the data sets to be more simply computed by comparing the three wild-type values to the three mutant values to identify significant changes.

An example analogy would be in the measurement of the length and weight of a cable. The three length measurements would be averaged, then the three weight measurements, then a single ratio determined that includes the variance (SD) of the two averages. For multiplication or division of numbers that contain known error (to establish fold changes, for example), a coefficient of variation (CV) should be calculated, squared, and used for computing the resulting CV of the ratio. That CV should then be converted to an SD for the final reported number. All of this being said, the resulting numbers do not change by a lot in the case of the presented data because the CVs of the data sets were comparable (e.g., the fold change data for *gldD* is reported as 6.25 +/- 2.87, which would change to 5.85 +/- 3.05). Nonetheless, a correction of the data would be more accurate, allow proper t-tests to be computed, and also contain fewer operations on the spreadsheets.

Answer: We recalculated the values as suggested by the reviewer (see the supplemental file). The new values are shown in the corresponding tables.

Worthan et al_ Answers to reviewer's comments

Minor issues:

In Figure 1, can the panel be flipped such that 'upper' is on the top and 'lower' is on the bottom? (or reverse the terminology, maybe use proximal and distal relative to the PTC). Not an issue, but considering the conservation of K90 as a histidine in eukaryotes, that would also be an interesting mutant to evaluate in bacteria.

Answer: We have made the suggested changes in Figure 1 to better illustrate the ribosome tunnel. Investigating the effects of uL22(K90H) substitution is a great idea but is currently beyond the focus of this manuscript.

For Figure 2, The doubling time is too slow for a healthy *E. coli* in LB at 37 C, suggesting either that the parental strain was compromised or that air or some nutrient was limiting. How was the growth rate calculated?

Answer: We have calculated these values using the times indicated in the figure and a non-linear regression formula (exponential Malthusian growth model, or simple exponential growth). However, considering that Reviewer #1 had expressed similar concerns, we redid these calculations with additional data using standard methods (linear methods with logarithmical values) as indicated in the above answer to reviewer 1. This time the doubling time values were around the 35 min range, still somewhat slow. As this reviewer pointed out, our strain contains several deletions used to analyze the expression of the reporter gene *tnaA-lacZ*. We suspect that the parental strain, which also show similar growth values, could have some growth limitations because it is missing some metabolic genes.

The time window is reported, but no log2 was calculated as far as I can tell. Why was the Klett data plotted by converting the Y-axis scaling instead of converting the data to log2 and fitting a line?

Answer: We have modified the plot as the reviewer suggested.

Why were the growth rates calculated from 160-260 minutes? This region overlaps with the observed growth rate reduction by the mutant. Perhaps an analysis both pre- and post-transition would better reflect the impact of the mutation on growth rate reduction. Unfortunately, the Klett values early in the growth curves are very noisy until ~20-32 units after 200 mins of culturing, which is generally past exponential phase under these conditions.

Answer: We have followed the reviewers advise and we have moved our time points from 160-260 to 140-240, which show linearity ($r=0.96$) under log2 values (see supplemental file).

Figure 4, the text and data are very hard to see. Can the important text and plot region sizes be increased?

Answers: We have increased the size of essential text as suggested

Figure 5A, the legend indicates the GAD assay was performed multiple times, but there are no values, averages, or SDs reported.

Answer: We have added values (new plot, Figure 5B) and calculations in the supplemental file and methods of calculations employed in the materials and methods.

Fig. 5C, the legend states that there were 3 cultures with indole and 4 without, but the plotted data shows 4 and 5 data points respectively. The survival clusters are ~100 fold different between the wild-type and mutant in each case, why is the p-value so high? Off hand, it seems you may have a 2 or 3 * P-value in each. How was the t-test performed and which data were used (the CFU or the resulting percentages)?

Answers: Our data needed more calculations. We have made additional experiments, resulting in a new Figure 5C. The new data and calculations are shown in the supplemental file.

Fig. 6 and related results. For Figure S2, how were the identities of the indicated spots verified as TnaA and Cdd?

Answer: Mass spectrophotometry was performed by Applied Biomics (Hayward, CA) to identify the proteins of the selected spots and indicated such in the corresponding supplementary data (See Fig S2A)

Worthan et al_ Answers to reviewer's comments

The RNAseq data indicated that *tnaA* expression was lowered ~2 fold, but the band for TnaA protein is nearly absent compared to the parental strain in the 2D gel (yet reported as 59%). In either case, why is it asserted repeatedly that this mutant strain cannot produce indole or that there are indole deficiencies?

Answer: We regularly measured the amount of indole in cultures using Kovacs reagents over the course of bacterial growth. We consistently observed lower accumulation of indole in the uL22 mutant strain. We had included this data to the supplementary data (Figure S2D) and provided an explanation in the main text (page 6, lines 30-39). It is important to point out that when TnaA expression is high, the amount of Trp is reduced and as a consequence the induction of the *tna* operon is reduced. We believe that the modest fold difference observed in *tnaA* transcripts (in RNA-seq) between the uL22 wild type and uL22(K90D) strains is the result of a reduction of Trp after high production of TnaA during cell growth in the wild type uL22-containing strain (please see Figure S2). This phenomenon had been previously proposed by Yanofsky et al (J.Bacteriol. 1991. 173:6009). We have added an explanation in the main text (Page 6, lines 25-42).

TnaA converts imported tryptophan to indole and it requires TnaB to do so. If the TnaB levels are the same, then the indole production is expected to be the same. Your medium provided tryptophan, was it consumed?

Answer: We are not sure about the concentration of Trp in the media; however, as we indicated in our previous answer the reduction of indole formation could signify that Trp is being consumed more in the cultures of the strains with uL22 wild type proteins.

The methods section could benefit from general editing. As examples, "hydrochloride acid" should be hydrochloric acid, "RPMs" should be RPM, and "L-glutamic acid potassium salt monohydrate" should be potassium glutamate, as it is dissolved.

Answer: We have edited the Materials and Methods as suggested.

Line numbering helps with commenting.

Answer: We have added line numbering this time for the benefit of the reviewers.

February 22, 2022

Dr. Luis Rogelio Cruz-Vera
University of Alabama in Huntsville
Biological Sciences
301 Sparkman
Huntsville, AL 35899

Re: Spectrum02261-21R1 (The identity of the constriction region of the ribosomal exit tunnel is important to maintain gene expression in *Escherichia coli*)

Dear Dr. Luis Rogelio Cruz-Vera:

Your manuscript has been accepted, and I am forwarding it to the ASM Journals Department for publication. You will be notified when your proofs are ready to be viewed.

Sincerely,

Amanda Oglesby
Editor, Microbiology Spectrum

Journals Department
Supplemental Material: Accept
Supplemental Material file 2: Accept